# PGO-BEN: Proxy-Guided Orthogonalization and Beta Ensembling for Few-Shot Domain-Incremental Learning

**Samrat Mukherjee**                                         *23d1599@iitb.ac.in*
*Centre for Machine Intelligence and Data Science, Indian Institute of Technology Bombay*

**Thivyanth Venkateswaran**                                  *thivyanth@iitb.ac.in*
*Department of Engineering Physics, Indian Institute of Technology Bombay*

**Eric Nuertey Coleman**                                     *eric.coleman@phd.unipi.it*
*Computer Science Department, University of Pisa*

**Luigi Quarantiello**                                       *luigi.quarantiello@phd.unipi.it*
*Computer Science Department, University of Pisa*

**Julio Hurtado**                                            *julio.hurtado@warwick.ac.uk*
*Centre for Applications of Mathematical & Computing Sciences, University of Warwick*

**Vincenzo Lomonaco**                                        *vlomonaco@luiss.it*
*Department of AI, Data and Decision Sciences, LUISS*

**Gemma Roig**                                               *roignoguera@em.uni-frankfurt.de*
*Department of Computer Science, Goethe University, Frankfurt*
*The Hessian Center for Artificial Intelligence (hessian.AI), Darmstadt, Germany*

**Subhasis Chaudhuri**                                       *sc@ee.iitb.ac.in*
*Department of Electrical Engineering, Indian Institute of Technology Bombay*

**Biplab Banerjee**                                          *bbanerjee@iitb.ac.in*
*Centre for Machine Intelligence and Data Science, Indian Institute of Technology Bombay*

**Reviewed on OpenReview:** *https://openreview.net/forum?id=jlb27FbHLv*

## Abstract

Continual adaptation to evolving domains with minimal supervision is essential for real-world deployment of machine learning systems. We formalize this objective as **Few-Shot Domain-Incremental Learning (FSDIL)**, where a model must adapt to each new domain using only a few labeled samples while retaining prior knowledge without access to previous data. This setting mirrors practical constraints in domains such as autonomous driving and medical imaging, where annotations are expensive and data retention is restricted by privacy regulations. Pre-trained vision–language models such as CLIP provide a strong initialization for FSDIL due to their transferable multi-modal representations. However, adapting CLIP incrementally under domain shifts remains challenging: few-shot updates often trigger *catastrophic forgetting* and insufficient *plasticity* across evolving distributions. To address these challenges, we introduce PGO-BEn (*Proxy-Guided Orthogonalization and Beta Ensembling*)—a rehearsal-free framework that leverages CLIP's semantic priors via prompt learning while preserving prior domain knowledge through two key mechanisms. (1) **Proxy-Guided Orthogonalization**(PGO): identifies conflicts between current gradients and proxy representations of past knowledge, inferred from current samples, and projects conflicting updates into an orthogonal subspace to prevent knowledge degradation. (2) **Beta Ensembling** (BEn): introduces a Beta-function-based temporal ensembling strategy

that adaptively balances stability and plasticity, outperforming conventional exponential moving average (EMA) approaches in retaining early-domain knowledge. We extensively evaluate PGO-BEN on three diverse benchmarks—**DomainNet**, **CoRE50**, and **CDDB-Hard**—and demonstrate consistent improvements over state-of-the-art domain-incremental and few-shot learning methods across all supervision levels in this challenging setting. Code: https://github.com/tarmas99/PGO-BEn

# 1 Introduction

Large-scale annotated datasets have catalyzed major advances in machine learning. However, collecting such datasets remains expensive, privacy-sensitive, and logistically challenging across many domains, especially those involving dynamic environments or regulated data (e.g., autonomous driving or clinical imaging). These limitations have spurred growing interest in data-efficient learning paradigms such as semi-supervised learning (SSL) van Engelen & Hoos (2019); Yang et al. (2022) and few-shot learning (FSL) Ravi & Larochelle (2017); Song et al. (2023), which reduce reliance on exhaustive manual annotation. Yet, these paradigms are typically designed for static data distributions and struggle in scenarios where models must continuously adapt to evolving environments, necessitating the study of *continual learning* (CL) Castro et al. (2018); Rebuffi et al. (2017); Riemer et al. (2019); Wang et al. (2022a).

Within CL, three principal paradigms have emerged: class-incremental learning (CIL), where new classes are introduced over time; task-incremental learning (TIL), where tasks differ and are explicitly identified; and domain-incremental learning (DIL), where input distributions shift across episodes while the label space remains constant. In this paper, we focus on DIL but challenge a common assumption in the literature—that each incoming domain offers abundant labeled data—overlooking the realistic scenario where new domains often provide only limited supervision due to data collection cost or privacy restrictions. In practical settings such as autonomous driving, robotic vision, and clinical imaging, models encounter sequential domain shifts (e.g., changes in lighting, device, or geography) but only limited labeled data per domain due to high annotation costs. In healthcare, for example, shifts across hospitals or scanners necessitate continual adaptation, yet privacy constraints prevent storing past data, while annotation costs restrict label availability Zhou et al. (2021). Likewise, deepfake detection requires identifying continually evolving fake content, and anomaly detection demands recognizing rare events while preserving knowledge of previously encountered anomalies. These scenarios motivate studying DIL under minimal supervision—learning from continually evolving domains with a fixed label space—yet this problem has been largely overlooked in existing literature.

Table 1: Comparison of continual learning settings. Our proposed setting (**FSDIL**) is highlighted.

| Setting | Label Space | Domain Shift | Label Budget in Incremental Session | Task/Domain ID Available |
|---|---|---|---|---|
| Class-Incremental Learning (CIL) | Expanding | No | Full supervision | No |
| Few-Shot Class-Incremental Learning (FSCIL) | Expanding | No | Few-shot | No |
| Domain-Incremental Learning (DIL) | Fixed | Yes | Full supervision | No |
| **Few-Shot Domain-Incremental Learning (FSDIL)** | **Fixed** | **Yes** | **Few-shot** | **No** |

To bridge the gap between current benchmarks and real-world applications, we introduce **Few-Shot Domain-Incremental Learning (FSDIL)** (Fig. 2a). In FSDIL, a model is first trained on a well-annotated base domain and must continually adapt to a stream of novel domains, each offering only a few labeled examples per class. FSDIL fundamentally differs from related paradigms (Table 1): unlike FSCIL Dong et al. (2021); Tao et al. (2020); Sur et al. (2025), it assumes a fixed label space with shifting domains; unlike Few-Shot Domain Adaptation (FSDA) Zhao et al. (2021), it requires continual rather than one-time adaptation; and unlike unsupervised DIL variants Mukherjee et al. (2025); Rakshit et al. (2022), it leverages limited supervision in each new domain, better reflecting real-world constraints. The base session involves training with abundant data, mirroring the common industrial practice of pre-training on large datasets. To respect privacy constraints in applications such as clinical imaging, our proposed FSDIL setting prohibits storing exemplars from past sessions.

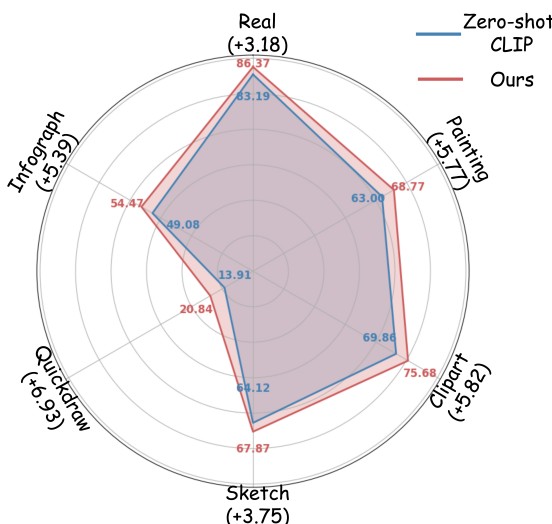

Figure 1: Comparison of Zero-shot CLIP-ViT-B/16 with PGO-BEN model (1-shot) across all domains in the DomainNet dataset. Numbers in brackets indicate the performance gain of our method.

Although large-scale models like CLIP exhibit strong generalization, they are insufficient for all domains. We evaluate the frozen CLIP-ViT/B-16 model Radford et al. (2021) using the prompt `"a photo of a --"` on the DomainNet dataset Peng et al. (2019), and compare it with our model (Fig. 1). The consistent performance gap across domains highlights that large-scale pretrained models fail to adapt effectively to domain shifts, necessitating specialized strategies for continual adaptation.

FSDIL is particularly challenging for three reasons. First, supervision is extremely limited: each new domain provides only a few labeled samples, making standard optimization prone to overfitting and variance collapse. Existing DIL methods like S-Prompt Wang et al. (2022a) and CP-Prompt Feng et al. (2024) fail under this regime, as their KNN-based domain prompt selection becomes biased with scarce data, leading to poor generalization. Second, sequential domain shifts occur without task boundary annotations, rendering exemplar-based or task-specific projection methods infeasible due to privacy, memory, or latency constraints. Third, large stylistic shifts across domains (e.g., photo → sketch) demand representations that remain label-consistent yet robust to visual variations.

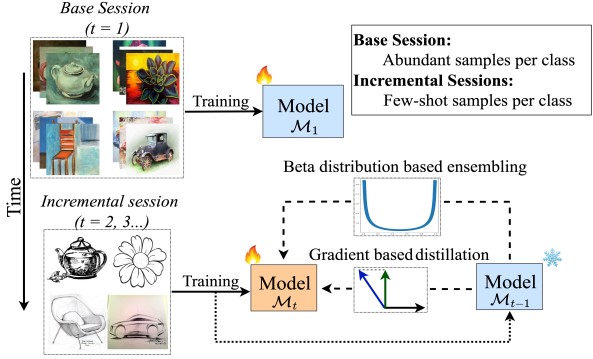
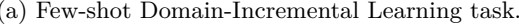

(a) Few-shot Domain-Incremental Learning task.

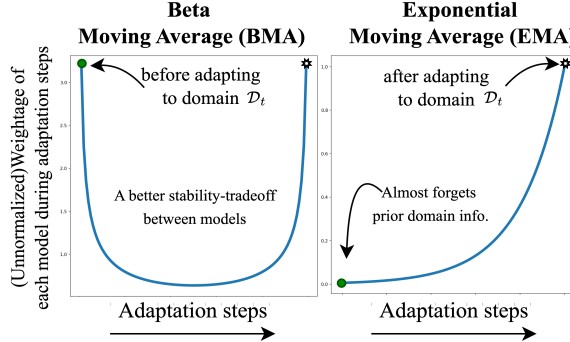

(b) Proposed $\beta$-function-based temporal ensembling for CL to improve knowledge retention.

Figure 2: (a) **FSDIL task.** A well-labeled base domain is followed by sparsely labeled incremental domains. Combined domain shift and label sparsity risk overfitting and forgetting. (b) **Beta Moving Average.** BMA uses a Beta distribution to adaptively weight model states, retaining prior domain knowledge, whereas EMA overly discounts earlier states and risks forgetting prior knowledge.

Given this setting, our work investigates three key questions: (Q1) *How to design a data-efficient adaptation strategy that generalizes well to new domains despite extreme supervision sparsity?* (Q2) *How can we mitigate catastrophic forgetting in a streaming setup without relying on task-specific gradient estimation or data replay?* (Q3) *Can we build a unified, robust representation that generalizes across diverse domains continually?*

**Our solution.** We propose PGO-BEN, a unified framework for FSDIL grounded in the semantic strength of the pre-trained vision–language model CLIP Radford et al. (2021), motivated by its recent success in DIL Feng et al. (2024); Wang et al. (2022a). The framework introduces three key innovations.

(1) **Efficient adaptation from few samples:** leveraging CLIP's few-shot capacity, we design a *multi-modal prompting* strategy that departs from prior prompt-tuning methods Zhou et al. (2022b;a); Wang et al. (2022a); Feng et al. (2024). Learnable prompts are injected across layers in both CLIP encoders, with text prompts conditioned on visual prompts to effectively capture domain shifts from minimal supervision while exploiting CLIP's prior knowledge. (2) **Mitigating catastrophic forgetting:** we introduce *proxy-guided orthogonalization*, aligning update directions with prior knowledge via cosine-based filtering. Unlike subspace projection methods Farajtabar et al. (2020); Liang & Li (2024); Lin et al. (2022); Saha et al. (2021), it requires no storage or approximation of past gradients—crucial when only few labeled samples are available—thus improving scalability as domains evolve. To further stabilize learning, a *Beta-function-based Moving Average (BMA)* replaces the standard EMA Caron et al. (2021); Carta et al. (2023), adaptively weighting model states through a symmetric Beta distribution to preserve early domain knowledge often lost with EMA (Fig. 2b). (3) **Learning domain-generalized representations:** moving beyond prompt-pool methods Feng et al. (2024); Wang et al. (2022a;c) that rely on domain-specific selection at inference, we condition text prompts on vision prompt tokens across layers. This enables dynamic adaptation to evolving visual styles, yielding domain-agnostic prompting and improved generalization without inference-time selection. Collectively, these designs make PGO-BEN an inference-efficient and scalable FSDIL framework—free from prompt memory, task identifiers, or exemplar buffers. In summary, this paper:

- Formalizes **FSDIL** as a realistic continual learning problem involving sequential domain shifts with few-shot supervision, while enforcing privacy-aware constraints by avoiding storage of past data.

- Proposes PGO-BEN, integrating multi-modal prompting, gradient-aligned distillation, and Beta-based ensembling to improve the stability–plasticity trade-off where domains evolve but the label space remains fixed.

- Demonstrates state-of-the-art performance on three benchmarks—**DomainNet** Peng et al. (2019), **CoRE50** Lomonaco & Maltoni (2017), and **CDDB-Hard** Li et al. (2023)—with supporting ablations.

## 2 Related Works

**Prompt Learning in CLIP.** Prompt learning has emerged as a lightweight alternative to full fine-tuning, originally developed for NLP Lester et al. (2021); Li & Liang (2021); Mishra et al. (2023) and later extended to vision-language models like CLIP Radford et al. (2021). In this context, CoOp Zhou et al. (2022b) introduces learnable prompts, while CoCoOp Zhou et al. (2022a) makes them input-conditioned to improve generalization. MaPLe Khattak et al. (2022) enriches CLIP's features by injecting hierarchical modality-aware prompts, and StyLIP Bose et al. (2024) further integrates domain-specific cues. We introduce learnable prompts across encoders, conditioning text prompts on vision prompts to more effectively capture and adapt to visual domain shifts.

**Data-Efficient Continual Learning.** CL aims to balance stability-plasticity tradeoff Wang et al. (2024); Zhou et al. (2024). Regularization-based methods like EWC Kirkpatrick et al. (2017), replay-based strategies like iCaRL Rebuffi et al. (2017) and ER Riemer et al. (2019), and parameter-isolation techniques Wang et al. (2023) address this trade-off with varying overhead. In domain-incremental learning, where task IDs are absent, GEM Lopez-Paz & Ranzato (2017) and latent-replay Pellegrini et al. (2020) help adapt to distribution shifts but rely on exemplar memory. Prompt-based continual learning methods such as S-Prompt Wang et al. (2022a), CP-Prompt Feng et al. (2024) learn per-domain prompts and rely on domain-aware retrieval during inference, requiring abundant labels and prompt selection mechanisms that increase latency—making

inference slower, quality that hinders application of FSDIL in real-world use cases. In contrast, FSDIL demands a unified, domain-agnostic prompting strategy that adapts continuously without task-specific routing or per-domain memory. Its more restrictive setting—lacking domain labels and extensive supervision—calls for lightweight, generalizable prompting across evolving domains.

**Gradient Projection in Continual Learning.** To mitigate forgetting, methods such as OGD Farajtabar et al. (2020), TRGP Lin et al. (2022), GPM Saha et al. (2021), and DualGPM Liang & Li (2023) constrain updates by projecting gradients onto orthogonal subspaces of prior tasks. While these leverage low-rank structures in gradient space, they require storing or estimating task-specific subspaces—an increasingly impractical demand as task count grows and data becomes scarce, as in FSDIL. In contrast, FSDIL necessitates scalable, memory-efficient approaches that preserve knowledge without explicit gradient storage or subspace tracking.

## 3 Proposed Methodology

**Problem Definition.** Consider a sequence of $\mathcal{N}$ distinct domains, $\{\mathcal{D}_1, \mathcal{D}_2, \cdots, \mathcal{D}_{\mathcal{N}}\}$, where each domain $\mathcal{D}_t$ during the training session $t$ consists of tuples $\{x_i^t, y_i^t\}_{i=1}^{|\mathcal{D}_t|}$. Here, $x_i^t$ represents an image, and $y_i^t$ denotes the corresponding label, with $y_i^t \in \mathcal{C} = \{c_1, c_2, \cdots, c_{|\mathcal{C}|}\}$. In particular, the set of classes $\mathcal{C}$ are identified from the beginning and will not change along the incremental learning stream Feng et al. (2024); Wang et al. (2022a), with $|\mathcal{C}|$ denoting number of distinct classes, and $|\mathcal{D}_t|$ denoting number of samples in domain $\mathcal{D}_t$.

In the FSDIL setting, the initial domain $\mathcal{D}_1$ is characterized by a large number of training samples per class within the label space $\mathcal{C}$. However, for each subsequent session $t > 1$, the number of training samples is significantly less, following the inequality $|\mathcal{D}_1| \gg |\mathcal{D}_t|$ for all $t > 1$. Specifically, we have $\forall t > 1, |\mathcal{D}_t| = |\mathcal{C}| \times n$, where $n$ is the number of samples per class in $\mathcal{C}$.

During each incremental session, training is restricted only to samples from the current domain $\mathcal{D}_t$, with data from previous domains unavailable for reuse, adhering to an exemplar-free setup Wang et al. (2022a). We evaluate the model after every session, where in any given session $t$, the model's performance is tested across all domains encountered up to that point. The main objective is to train a model that adapts to each new domain with few labeled examples while retaining knowledge from previously seen domains.

### 3.1 PGO-BEN for Few-Shot Domain-Incremental Learning

To address the constraints of FSDIL— generalizing with minimal supervision, adapting to distributional shifts across domains, and absence of task boundaries—we build on the generalization capabilities of large-scale vision-language models. PGO-BEN is designed to exploit vision-language priors of CLIP-like models for FSDIL. Specifically, we leverage CLIP as the backbone for our proposed model $\mathcal{M}$, formally defined as $\mathcal{M} = \{\mathcal{F}_T, \mathcal{F}_V, \mathcal{P}, Pr, \text{TOK}_T, \text{TOK}_V\}$, where $\mathcal{F}_T$ and $\mathcal{F}_V$ are the frozen text and vision encoders of CLIP, respectively. $\mathcal{P} = \{\mathcal{P}^1, \cdots, \mathcal{P}^J\}$ denotes the Encoder-Synergy module, a set of lightweight projector networks that modulate learnable tokens $\text{TOK}_T$ and $\text{TOK}_V$ at $J$ intermediate layers in Text and Vision encoder respectively. To ensure semantic alignment of the text encoder representations under changing visual domains, we condition $\text{TOK}_T$ on corresponding $\text{TOK}_V$ via Encoder-Synergy module, enabling the text encoder to adapt to evolving visual domains and support domain-invariant representation learning. The unified learnable prompt representation $Pr = [v_1][v_2] \cdots [v_m][\text{CLS}]$, where $[v_i]$'s are learnable tokens and $[\text{CLS}]$ denotes the classification token, serves as input to $\mathcal{F}_T$, avoiding manual domain-specific prompt tuning.

At each incremental session $t$, the model adapts to domain $\mathcal{D}_t$ with updated parameters $\theta_t = \{\mathcal{P}_t, Pr_t, \text{TOK}_{T_t}, \text{TOK}_{V_t}\}$. To effectively address FSDIL challenges, PGO-BEN comprises three synergistic components: (i) a multi-modal prompting strategy that enables unified adaptation across visual and textual branches (Sec. 3.1.1); (ii) A proxy-guided orthogonal gradient update strategy, where the model trained till session $t-1$, $\mathcal{M}_{t-1}$ serves as a proxy for prior domains, ensuring that gradient updates for the current domain remain aligned to previously learned knowledge and do not cause forgetting. (Sec. 3.1.2); and (iii) a Beta function-based temporal ensembling strategy that adaptively ensembles model states over time to enhance knowledge retention under domain shift without hampering the plasticity of the model (Sec. 3.1.3). Together, these components allow PGO-BEN to retain prior knowledge while flexibly adapting

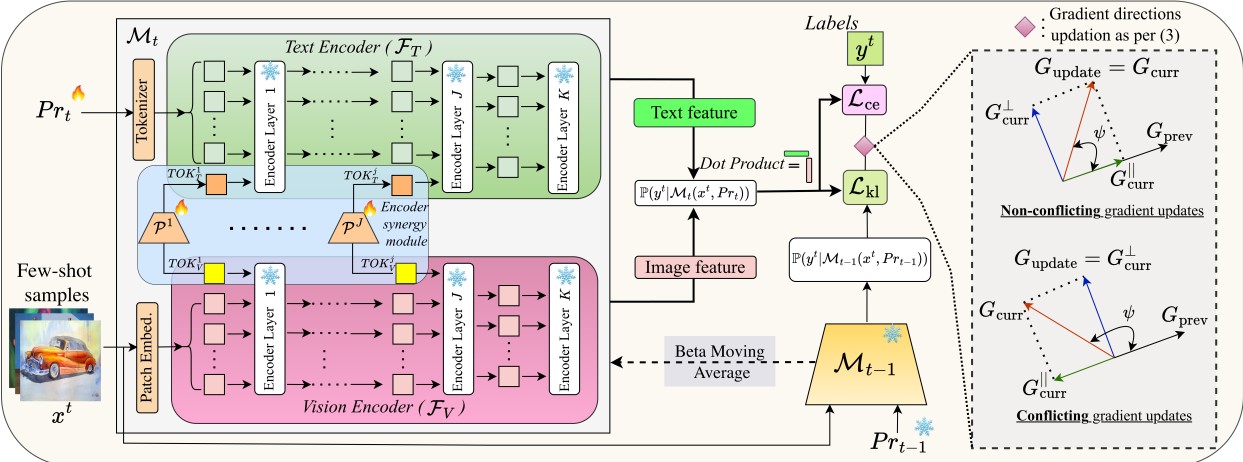

Figure 3: **Overview**: PGO-BEN uses a multi-modal prompt learning strategy to learn a generalizable representation from few-shot examples observed in domain $\mathcal{D}_t$. To adapt to the current incremental domain $\mathcal{D}_t, \forall t > 1$, PGO-BEN uses the labeled examples of current domain data guided by cross-entropy loss $\mathcal{L}_{\text{ce}}$. PGO-BEN compares the prediction of current model $\mathcal{M}_t$ and previous model $\mathcal{M}_{t-1}$ on the current data $\mathcal{D}_t$ using $\mathcal{L}_{\text{kl}}$. PGO-BEN introduces an adaptive gradient direction selection strategy to mitigate forgetting of knowledge of past domains guided by the proxy-knowledge of the past domains from $\mathcal{L}_{\text{kl}}$. Since $\mathcal{M}_{t-1}$ has never seen $\mathcal{D}_t$, Beta-Moving Average ensembling technique is used to further enhance the stability-plasticity trade-off, while mitigating the risk of unreliable updates. Here $\mathbb{P}(y|\cdot)$ denotes posterior prediction probability.

to new domains from limited samples. All learnable parameters are trained to obtain a domain-agnostic representation applicable for all seen domains, avoiding any domain-specific parameter selection steps prevalent in DIL literature Wang et al. (2022a); Feng et al. (2024) Fig. 3 illustrates the process.

### 3.1.1 Multi-Modal Prompting for Domain-Invariant Representation Learning

Conventional prompt tuning for CLIP, such as CoOp and CoCoOp Zhou et al. (2022b;a), adapts only the text encoder, offering limited robustness to visual domain shifts. MaPLe Khattak et al. (2022) enriches vision features with text representations, failing to capture evolving visual distributions—an essential requirement in FSDIL, where supervision is sparse and domain relations across incremental sessions are undefined.

We propose a simple *multi-modal prompting* strategy, specifically for FSDIL, that conditions text tokens on visual cues across layers. Specifically, each transformer block $j$ in CLIP's encoders includes learnable tokens $\text{TOK}_V^j$ and $\text{TOK}_T^j$, linked via a layer-specific lightweight projector $\mathcal{P}^j$ such that:

$$\text{TOK}_T^j = \mathcal{P}^j(\text{TOK}_V^j)$$

Since the feature representations evolve across transformer layers, we introduce layer-specific projectors rather than a single shared projector, allowing flexible cross-modal alignment at different semantic depths. We conducted an ablation study on the number of layers the projector is needed to be introduced in **Sup.Mat.**. By propagating visual-domain cues into the text encoder, the model dynamically aligns its textual embedding space with evolving visual distributions, thereby enhancing domain invariance without explicit supervision (see Fig. 4a). We initialize this unified prompt space by training on the labeled base domain $\mathcal{D}_1$ using cross-entropy loss, enabling the model to encode domain-agnostic semantics into the prompt tokens, supporting stable adaptation across subsequent domains with minimal supervision. Table 8 & Fig. 4a highlight the superiority of conditioning text prompts on visual tokens which enables better adaptation to changing visual domains compared to other conditioning techniques Khattak et al. (2022) and baseline methods.

### 3.1.2 Retain While You Learn: Proxy-Guided Orthogonalization for Stability under Shift

Prior works Kirkpatrick et al. (2017); Saha et al. (2021); Lin et al. (2022) attribute catastrophic forgetting in CL to gradient updates of the current task which overrides the knowledge of the past tasks in order to learn

the current task . Prior methods Liang & Li (2024); Saha et al. (2021); Lin et al. (2022); Liang & Li (2023) address forgetting by projecting current task gradient updates orthogonally to a pre-computed approximation of the gradient subspaces of the previous tasks. Such orthogonalization mitigates the risk of interference of knowledge of the past domains with gradient update of the current domain. However, approximating past gradient spaces scales poorly with an increasing number of tasks and require memory overhead Liang & Li (2024); Lin et al. (2022). Under limited supervision constraint of FSDIL, such approximations become unreliable, leading to degraded performance. Motivated by Yu et al. (2020), we propose a memory-free strategy that mitigates catastrophic-forgetting by deriving a proxy representation of prior domain knowledge from the few-shot samples of the current domain. This proxy knowledge is used to regularize the learning dynamics of the current domain, to preserve knowledge of previously seen domains without the need for explicitly storing pre-computed gradient spaces or approximating gradients of past domains from limited data, unlike prior methods Liang & Li (2023); Saha et al. (2021); Liang & Li (2024).

At session $t > 1$, we initialize the current model $\mathcal{M}_t$ and prompt $Pr_t$ from the previous model $\mathcal{M}_{t-1}$, which has learned from $\{\mathcal{D}_1, \ldots, \mathcal{D}_{t-1}\}$. This provides an enriched initialization as compared to random initialization for every domain as in Wang et al. (2022a); Feng et al. (2024). While adapting $\mathcal{M}_t$ to domain $\mathcal{D}_t = \{(x_i^t, y_i^t)\}_{i=1}^{|\mathcal{D}|_t}$, we use the frozen $\mathcal{M}_{t-1}$ model as a functional proxy of knowledge of previously seen domains to regulate the direction of parameter updates. For each input $x_i^t$, we compute:

- **Cross-Entropy Loss** for adaptation: $\mathcal{L}_{\text{ce}}(\hat{y}_i^t, y_i^t)$, where $\hat{y}_i^t = \mathcal{M}_t(x_i^t, Pr_t)$

- **KL Divergence** to preserve past knowledge:

$$\mathcal{L}_{\text{kl}} = -\sum_i \mathcal{M}_{t-1}(x_i^t, Pr_{t-1}) \log \frac{\mathcal{M}_t(x_i^t, Pr_t)}{\mathcal{M}_{t-1}(x_i^t, Pr_{t-1})} \tag{1}$$

We compute gradients $G_{\text{curr}} = \nabla_{\theta_t} \mathcal{L}_{\text{ce}}$ and $G_{\text{prev}} = \nabla_{\theta_{t-1}} \mathcal{L}_{\text{kl}}$, and calculate the angle between them:

$$\psi = \cos^{-1}\left(\frac{G_{\text{prev}} \cdot G_{\text{curr}}}{\|G_{\text{prev}}\| \|G_{\text{curr}}\|}\right) \tag{2}$$

The angle $\psi$ indicates whether the gradient update for adapting to the current domain is consistent with, or conflicting against, the model's existing knowledge of previously seen domains. If the existing knowledge of the model align with the update direction ($\psi \in [-\frac{\pi}{2}, \frac{\pi}{2}]$), we hypothesize that updating the model following the update gradient $G_{\text{curr}}$ will not cause any forgetting (as, $G_{\text{curr}}^{\|}$ is along the direction of $G_{\text{prev}}$, see Fig. 3).

In case of conflicting alignments(i.e. $\psi \notin [-\frac{\pi}{2}, \frac{\pi}{2}]$), updating the model parameters with $G_{\text{curr}}$ will adapt the model on current domain $\mathcal{D}_t$ at the cost of forgetting the knowledge of the previously seen domains. In such scenario, we project $G_{\text{curr}}$ onto the orthogonal space of $G_{\text{prev}}$ to obtain $G_{\text{curr}}^{\perp}$, and update the model using,

$$G_{\text{update}} = \begin{cases} G_{\text{curr}}, & \psi \in [-\frac{\pi}{2}, \frac{\pi}{2}] \\ G_{\text{curr}}^{\perp}, & \text{otherwise} \end{cases} \quad, \theta_t \leftarrow \theta_t - \eta \cdot G_{\text{update}} \tag{3}$$

**Benefits of our alignment strategy:** Our adaptive-alignment strategy avoids storing the gradient-space information which has increasing memory overhead as we see more domains, we only maintain the prior model state $\mathcal{M}_{t-1}$, a very standard practice in knowledge-distillation literature. Unlike prior methods that constrain updates to fixed subspaces Liang & Li (2024); Lin et al. (2022), we retain past knowledge by dynamically adjusting update directions using $G_{\text{prev}}$ as a proxy knowledge of prior domains. Updating the parameters in the orthogonal space of the previous gradient directions in conflicting scenarios, reduces the risk of forgetting prior domain knowledge. This implicit and scalable alignment maintains plasticity without relying on unreliable gradient subspace approximations in low-data regimes.

### 3.1.3 From EMA to BMA: Temporal Ensembling that Remembers

Since the model $\mathcal{M}_{t-1}$ has never been trained on domain $\mathcal{D}_t$, relying solely on the gradient information maybe unreliable and could give sub-optimal performance in knowledge retention and may also hurt the

plasticity. To avoid the ill-effects of unreliable gradient updates, we use temporal ensembling of model states. In CL literature, EMA smoothing Carta et al. (2023); Caron et al. (2021) which is commonly used to stabilize updates, as we observe in Table 3, underperforms under large, unconstrained domain shifts typical in FSDIL, due to the fixed decay nature (Fig. 2b) .

We propose a more flexible *Beta Moving Average (BMA)* strategy that adaptively ensembles intermediate model states during training, based on the Beta distribution. Unlike EMA, which monotonically discounts early checkpoints, BMA assigns higher weight to both early and late training phases. Early model states retain knowledge about the previously seen domains, where as later model states have adapted to current domain, but have a risk of forgetting the prior knowledge. BMA improves stability and mitigating task-recency bias when under sparse labels and large domain shifts, better than EMA, thus being more suitable for FSDIL task, strengthening stability-plasticity dilemma (see Fig. 2b and Table. 3).

**Beta-function based Ensembling.** During $T'$ update steps on domain $\mathcal{D}_t$, the model $\mathcal{M}_t$, which is initialized with $\mathcal{M}_{t-1}$, has knowledge about $\{\mathcal{D}_1, \ldots, \mathcal{D}_{t-1}\}$ yet underfits $\mathcal{D}_t$. In later iterations, the model adapts to $\mathcal{D}_t$, risking forgetting of prior knowledge. For better stability-plasticity trade-off, we treat the final model $\mathcal{M}_t$ as a weighted ensemble of intermediate models $\{\mathcal{M}_{t'}\}_{t'=0}^{T'}$, using $\beta$-function based ensembling as:

$$\mathcal{M}_t = \sum_{t'=0}^{T'} \frac{\alpha_{t'}}{\sum_{k=0}^{T'} \alpha_k} \mathcal{M}_{t'}, \quad \text{where } \alpha_{t'} = \text{Beta}(\beta, \beta) \left( \frac{t' + 0.5}{T' + 1} \right). \tag{4}$$

With $\beta < 1$, BMA highlights both early and late model states, enhancing memory of past domains while incorporating the current one (see Fig. 2b).

**Efficient Online Implementation.** In order to minimize memory overhead, rather than storing every intermediate state, we implement BMA as a running average:

$$\mathcal{M}_{t'}^{BMA} = \frac{\sum_{k=0}^{t'-1} \alpha_k}{\sum_{k=0}^{t'} \alpha_k} \mathcal{M}_{t'-1}^{BMA} + \frac{\alpha_{t'}}{\sum_{k=0}^{t'} \alpha_k} \mathcal{M}_{t'}. \tag{5}$$

This low-memory update requires only a single auxiliary model state, offering temporal smoothing that complements gradient-based updates Shu et al. (2023). Each iteration of training applies proxy-guided orthogonal update followed by BMA integration to ensure stable adaptation across sessions.

During **inference**, we deploy the final BMA model $\mathcal{M}_{T'}^{BMA}$ for evaluations on all test samples across the domains $\{\mathcal{D}_1, \ldots, \mathcal{D}_t\}$ and initialize $\mathcal{M}_{t+1}$ for adaptation to domain $\mathcal{D}_{t+1}$ with $\mathcal{M}_{T'}^{BMA}$, and doesn't require any prompt-selection phase, resulting in efficient inference. Pseudocode are provided in ***Sup. Mat.***

## 3.2 On the Utility of PGO-BEN for FSDIL

To theoretically justify PGO-BEN in the FSDIL setting, we employ PAC-Bayesian theory McAllester (1999). For a model $\mathcal{M}_t \sim \rho$ (posterior over parameters $\theta_t$) adapted to domain $\mathcal{D}_t$ with $n$ examples, and a prior $\pi$ (e.g., CLIP-initialized $\theta_0$), the expected true risk $\mathcal{L}_{\mathcal{D}_t}(\mathcal{M}_t)$ is bounded with probability $\geq 1 - \delta$ by:

$$\mathbb{E}_{\mathcal{M}_t \sim \rho}[\mathcal{L}_{\mathcal{D}_t}(\mathcal{M}_t)] \leq \mathbb{E}_{\mathcal{M}_t \sim \rho}[\hat{\mathcal{L}}_t(\mathcal{M}_t)] + \sqrt{\frac{KL(\rho||\pi) + \log \frac{2\sqrt{n}}{\delta}}{2n}} \tag{6}$$

This bound links true risk to empirical risk $\hat{\mathcal{L}}_t(\mathcal{M}_t)$, the complexity term $KL(\rho||\pi)$, and sample size $n$. Contrastingly, the bound discussed in Shi & Wang (2023) relies on storing examples from previous domains in a memory buffer, contrasting our exemplar-free motivation. PGO-BEN's components aim to tighten this bound, achieving better adaptation to target domains:

**CLIP Initialization as an Informative Prior** $\pi$: CLIP initialization ensures $\pi$ is centered in a robust, generalizable region. For few-shot domains, the learned posterior $\rho$ needs minimal deviation from $\pi$ to minimize empirical risk, directly reducing the $KL(\rho||\pi)$ term and tightening the bound.

**Gradient-Aligned Distillation for Posterior Stability**: The gradient alignment mechanism (Sec 3.2.2) stabilizes $\rho$ by constraining updates based on prior knowledge ($\mathcal{M}_{t-1}$). This prevents drastic parameter shifts, keeping $\rho$ closer to $\pi$ and thus helping maintain a small $KL(\rho||\pi)$. This stability also limits sensitivity to few samples, reducing the empirical-to-expected risk gap and yielding a more reliable $\hat{\mathcal{L}}_t(\mathcal{M}_t)$.

**Beta Ensembling for Posterior Regularization**: BMA (Sec 3.2.3) defines $\rho$ as an implicit Beta-weighted mixture of intermediate model states. This averaging reduces estimator variance, leading to a more stable and potentially lower empirical risk $\mathbb{E}_{\mathcal{M}_t \sim \rho}[\hat{\mathcal{L}}_t(\mathcal{M}_t)]$. Such ensembling also regularizes $\rho$, potentially finding flatter minima associated with better generalization and favorably impacting the complexity term.

Further formal and empirical discussions in this regard are provided in ***Sup. Mat***.

## 4 Experimental Evaluations

**Datasets.** Following DIL literature Wang et al. (2022a); Feng et al. (2024), we evaluate our method on three DIL benchmarks: DomainNet Peng et al. (2019), CORe50 Lomonaco & Maltoni (2017), and CDDB-Hard Li et al. (2023). These datasets vary in both the number of classes and the domains the model must adapt to. We follow the domain adaptation order of Rakshit et al. (2022) for DomainNet and we follow the domain order for CDDB-Hard and CoRE50 detailed in Wang et al. (2022a); Feng et al. (2024). A detailed description of each dataset and the implementation details are provided in ***Sup. Mat.***

**Evaluation Metrics.** Following recent work Zhu et al. (2023); Bendou et al. (2025), we conduct few-shot experiments with $1, 2, 4$, and $8$ shots to assess the effectiveness of our approach. We measure performance using two standard metrics: (i) **Average Accuracy (AA)** Rakshit et al. (2022): The mean classification accuracy across all domains seen so far, and (ii) **Forgetting Alleviation (FA)** Liu et al. (2023): The mean accuracy on a domain after the model adapts to subsequent domains in the continual stream. We report the overall AA* and overall FA*—averaged across all sessions like Mukherjee et al. (2025). Detailed definitions of these metrics, per-domain results, are provided in ***Sup. Mat.*** Results are reported with average of three random seeds. Further results with five random seeds are present in ***Sup. Mat.***

**Baselines.** We compare our method to multiple baselines adapted to the FSDIL setting. For **regularization-based** approaches, we incorporate EWC Kirkpatrick et al. (2017) and LwF Li & Hoiem (2017) into our backbone, updating only the learnable prompt. We also consider **prompt-based DIL** methods such as L2P Wang et al. (2022c), S-Prompt Wang et al. (2022a), CP-Prompt Feng et al. (2024), and a **LoRA-based prior gradient approximation** technique, InfLORA Liang & Li (2024). We re-run all baselines under the same experimental setup on every dataset to ensure fair comparisons and report the average over three runs with three random seed values. In contrast to the baselines which require a prompt pool(separate set of prompts for each individual domains), our method learns and continually updates a fixed set of parameters, to avoid domain-prompt selection during inference prevalent in recently proposed parameter isolation based DIL strategies Wang et al. (2022a); Feng et al. (2024). Considering the closed-set nature of the FSDIL task, we choose not to compare our method against any FSCIL baselines. We used the following prompts for the Zero-shot CLIP experiments `a photo of a _`, `a photo of a _ image` and `there is a _ in this image.`, for DomainNet, CDDB-Hard and CoRE50 respectively, comparison with other prompts in ***Sup.Mat.***.

### 4.1 Experimental Results

We use CLIP-ViTB/16 as the backbone and re-run all baselines accordingly for fair comparison under DIL benchmarks. Wang et al. (2022a); Feng et al. (2024). Table 2 presents the average performance of PGO-BEN across 1-, 2-, 4-, and 8-shot settings on DomainNet, CDDB-Hard, and CoRE50 datasets (per-shot results in ***Sup. Mat.***). PGO-BEN consistently outperforms all baselines, including those that maintain domain-specific prompt pools for reducing forgetting and improving recognition. On DomainNet, PGO-BEN achieves a gain of +1.31% in AA* and +1.27% in FA* over the strongest baseline. For CDDB-Hard and CoRE50, it surpasses prompt-pool-based methods by +6.66% and +2.53% in FA*, and by +6.56% and +4.86% in AA*, respectively—demonstrating substantial improvements in balancing stability and plasticity. Importantly, PGO-BEN attains these results without relying on prompt pools or domain-specific routing at inference,

Table 2: **Comparison across DomainNet, CDDB-Hard, and CoRE50 averaged over 1, 2, 4, and 8-shot settings**. **Bold** and underlined denote the best and second-best scores. PGO-BEN outperforms all baselines without using prompt pools, demonstrating its generalization strength. * indicates CLIP-ViTB/16-based reimplementation. Results are mean ± std over 3 seeds. Red font denotes least std method.

| Method | Prompt Pool | Backbone | DomainNet Average AA*(↑) | DomainNet Average FA*(↑) | CDDB-Hard Average AA*(↑) | CDDB-Hard Average FA*(↑) | CoRE50 Average AA*(↑) | CoRE50 Average FA*(↑) |
|---|---|---|---|---|---|---|---|---|
| DyTox Douillard et al. (2021) | × | ViT | $30.11_{\pm0.90}$ | $19.02_{\pm0.64}$ | $57.17_{\pm0.60}$ | $53.17_{\pm0.82}$ | $46.57_{\pm0.83}$ | $28.72_{\pm0.95}$ |
| Zero-shot CLIP Radford et al. (2021) | × | CLIP | 69.05 | — | 56.32 | — | 12.67 | — |
| LwF* Li & Hoiem (2017) | × | CLIP | $72.06_{\pm0.82}$ | $60.70_{\pm0.93}$ | $68.25_{\pm0.74}$ | $58.99_{\pm0.75}$ | $64.41_{\pm0.78}$ | $57.75_{\pm0.60}$ |
| EwC* Kirkpatrick et al. (2017) | × | " | $70.92_{\pm0.90}$ | $58.85_{\pm0.89}$ | $71.30_{\pm0.81}$ | $62.21_{\pm0.77}$ | $63.41_{\pm0.68}$ | $55.60_{\pm0.44}$ |
| L2P* Wang et al. (2022c) | ✓ | " | $67.08_{\pm0.64}$ | $54.61_{\pm0.48}$ | $\underline{73.53}_{\pm0.94}$ | $65.81_{\pm0.83}$ | $79.88_{\pm0.76}$ | $78.36_{\pm0.92}$ |
| DualPrompt* Wang et al. (2022b) | ✓ | " | $73.45_{\pm0.66}$ | $63.50_{\pm0.76}$ | $73.08_{\pm0.66}$ | $66.51_{\pm0.80}$ | $55.61_{\pm0.50}$ | $50.54_{\pm0.71}$ |
| S-Prompt Wang et al. (2022a) | ✓ | " | $67.64_{\pm0.37}$ | $56.13_{\pm0.32}$ | $65.31_{\pm0.56}$ | $60.22_{\pm0.52}$ | $79.23_{\pm0.77}$ | $76.31_{\pm0.53}$ |
| CODA-Prompt Smith et al. (2023) | ✓ | " | $\underline{73.50}_{\pm0.80}$ | $\underline{63.85}_{\pm0.62}$ | $70.53_{\pm0.50}$ | $60.45_{\pm0.47}$ | $56.81_{\pm0.71}$ | $53.73_{\pm0.72}$ |
| InfLORA* Liang & Li (2024) | × | " | $71.93_{\pm0.59}$ | $60.41_{\pm0.76}$ | $66.65_{\pm0.48}$ | $56.65_{\pm0.77}$ | $65.30_{\pm0.90}$ | $58.10_{\pm0.72}$ |
| CP-Prompt Feng et al. (2024) | ✓ | " | $71.89_{\pm0.82}$ | $60.87_{\pm0.70}$ | $66.95_{\pm0.36}$ | $62.10_{\pm0.22}$ | $\underline{81.57}_{\pm0.64}$ | $\underline{79.99}_{\pm0.54}$ |
| PGO-BEN (Ours) | × | CLIP | $\mathbf{74.76}_{\pm0.17}$ | $\mathbf{64.77}_{\pm0.26}$ | $\mathbf{80.09}_{\pm0.29}$ | $\mathbf{73.17}_{\pm0.16}$ | $\mathbf{86.43}_{\pm0.35}$ | $\mathbf{82.52}_{\pm0.53}$ |
| | | Δ | +1.31 | +1.27 | +6.56 | +6.66 | +4.86 | +2.53 |

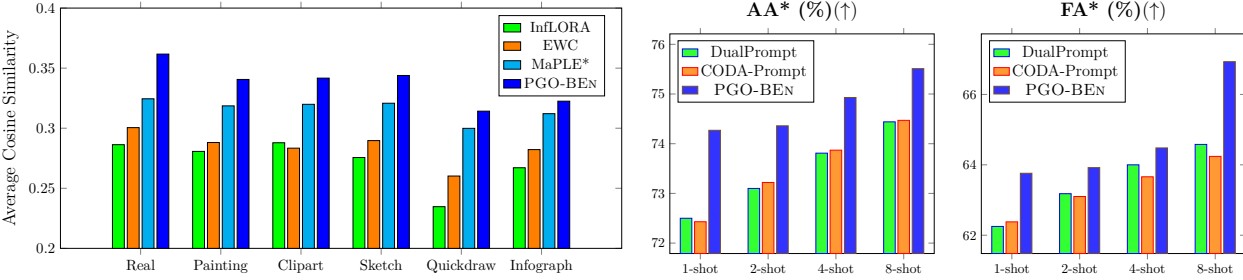

(a) Learnt representation of PGO-BEN is more aligned.  (b) PGO-BEN retains better with more data.

Figure 4: (a) **Encoder Representation Alignment.** We compare the representation similarity of the two encoder representations. Encoder representation of PGO-BEN have consistent higher cosine-similarity for all domains, indicating that Text-encoder is more aligned to the changing visual distribution in the Vision encoder as compared to other baselines. MaPLE* also fails to effectively capture the evolving visual distribution owing to a unsuitable encoder alignment direction. (b) **Sample Efficiency.** On DomainNet dataset, PGO-BEN consistently outperforms top-2 baselines across varying level of supervision.

offering a memory- and computation-efficient solution compared to methods that rely on prompt-pools (like Feng et al. (2024); Wang et al. (2022a), having a KNN based selection steps). Given that AA* and FA* aggregate performance over all sessions (*see **Sup. Mat.***), even moderate gains reflect meaningful improvements in continual learning dynamics. Our method achieves the lowest standard deviation across runs, indicating more consistent performance compared to all baseline methods.

These results highlight the effectiveness of our design: CLIP-based multi-modal prompting, with text-encoder prompt tokens ($\text{TOK}_T$) conditioned on image-encoder prompts ($\text{TOK}_V$) improves cross-domain adaptability with few labeled samples, while proxy-guided orthogonalization and Beta-function-based temporal ensembling enhance long-term retention across evolving domains, without continually increasing memory overhead.

**Representational alignment:** We compare the representation similarity of the two CLIP encoder representations by computing the average cosine similarity of the image embedding and text embedding of various baselines on the test dataset of the "Real" domain of the DomainNet dataset, as the models sequentially adapt to the new domains. As observed in Fig. 4a, PGO-BEN maintains consistently higher image–text embedding similarity than all baselines, indicating stronger representational alignment of and better retention under domain shift. In contrast, MaPLE*, which conditions vision prompts on text prompts, shows weaker alignment. These results highlight the effectiveness of our conditioning strategy in both aligning the text encoder to the evolving visual distribution and preserving stability across domains.

Table 3: **Efficacy of proposed BMA compared to EMA** on DomainNet dataset. BMA shows superiority in retaining prior knowledge. Results are mean over 3 seeds.

| Technique | 1-shot | | 2-shots | | 4-shots | | 8-shots | |
|---|---|---|---|---|---|---|---|---|
| | AA*↑ | FA*↑ | AA*↑ | FA*↑ | AA*↑ | FA*↑ | AA*↑ | FA*↑ |
| EMA ($\lambda = 0.98$) | 64.93 | 49.53 | 72.77 | 61.28 | 73.99 | 62.81 | 74.18 | 63.30 |
| EMA ($\lambda = 0.99$) | 58.22 | 45.77 | 66.10 | 53.99 | 71.58 | 59.73 | 73.34 | 61.90 |
| BMA ($\beta = (0.5, 0.5)$) | **74.27** | **63.76** | **74.36** | **63.92** | **74.53** | **64.17** | **74.61** | **64.28** |

Table 4: **Robustness analysis of BMA ($\beta = (0.5, 0.5)$) over EMA ($\lambda = 0.98$).** We compare the cosine-similarity of the encoder representation on the test data of "Real" domain after adapting to "Clipart" domain (left) and "Sketch" domain (right), for PGO-BEN with EMA and PGO-BEN with BMA. BMA variant is observed to be superior, thus highlighting that it is stable even when domain gaps are large. Results are mean±std over 3 seeds.

| | Real → Clipart | Real → Sketch |
|---|---|---|
| PGO-BEN with EMA ($\lambda = 0.98$) | $32.50_{\pm 0.23}$ | $31.95_{\pm 0.25}$ |
| PGO-BEN with BMA ($\beta = (0.5, 0.5)$) | $\mathbf{33.03_{\pm 0.21}}$ | $\mathbf{32.06_{\pm 0.27}}$ |
| $\Delta$ | +0.53 | +0.11 |

## 4.2 Ablation Analysis

**(a) Sensitivity to Training Sample Availability.** We analyze the effect of supervision sparsity by varying labeled samples per class in DomainNet (Fig. 4b). PGO-BEN consistently outperforms the top baselines across all supervision levels, achieving superior generalization and retention in low-shot regimes (e.g., +1.77% AA* and +1.38% FA* in 1-shot). This advantage further increases with more labels (e.g., +2.35% FA* over DualPrompt in 8-shot), confirming that PGO-BEN scales effectively with available supervision while maintaining cross-domain consistency.

**(b) Comparison of EMA and BMA.** We compare BMA with EMA across varying shots on DomainNet (Table 3). BMA consistently yields superior performance by more effectively balancing the stability–plasticity trade-off. While EMA applies exponentially decaying weights that rapidly diminish the influence of early training states, it often leads to aggressive forgetting of knowledge from prior domains. In contrast, BMA assigns symmetric Beta-function based weights (see Fig. 2b), preserving early domain knowledge while still adapting to the current task—resulting in improved retention. Moreover, EMA is highly sensitive to the decay hyperparameter, requiring careful tuning across domains. BMA exhibits greater robustness under hyperparameter variation (Table 6), further supporting its suitability for dynamic FSDIL settings.

To assess the retention benefits of BMA over EMA, we compute the cosine similarity between the final CLIP embeddings and those obtained after learning the "Real" domain, evaluated on "Real" test samples after adaptation to Clipart and Sketch. Higher similarity indicates better knowledge preservation. As shown in Tab. 4, under the 4-shot setting, the BMA variant of PGO-BEN consistently maintains higher similarity across both adaptation scenarios (Real → Clipart and Real → Sketch), demonstrating stronger stability against forgetting.

To further investigate the reasoning of such behavior in a continual adaptation scenario, in Tab. 5 , we compare the variance of the prediction vector of PGO-BEN with EMA and PGO-BEN with BMA on the test dataset of the "Real" domain of the DomainNet dataset, with respect to the prediction vector initially obtained after just training on the "Real" domain, as both the models sequentially adapt to the new domains. In the 4−shot scenario, we observe that EMA variant of the model has relatively higher variance compared to the BMA variant. This highlights why the performance of BMA remains more stable compared to EMA variant, highlighting the better stability thus achieved.

Clearly, these findings validate BMA as a more principled and stable ensembling strategy than EMA, enhancing long-term knowledge preservation in continual few-shot learning.

**(c) Sensitivity to $\beta$ in Beta Ensembling.** We evaluate the effect of different $\beta$ values in the Beta distribution used to weigh intermediate model states $\{\mathcal{M}_{t'}\}_{t'=0}^{T'}$ during adaptation to domain $\mathcal{D}_t$ (Table 6).

Table 5: **Comparison of prediction variance between EMA ($\lambda = 0.98$) and BMA($\beta = (0.5, 0.5)$) ensembling on "Real" domain test set during continual domain adaptation.** We compute the variance of the prediction logit vector of PGO-BEN with EMA and PGO-BEN with BMA, on all models which has adapted to the subsequent domains, on the test dataset of "Real" domain. PGO-BEN with BMA is observed to have **less** variance as compared to PGO-BEN with EMA. Results are mean±std over 3 seeds.

| | Painting | Clipart | Sketch | Quickdraw | Infograph |
|---|---|---|---|---|---|
| PGO-BEN with EMA ($\lambda = 0.98$) | $4.71_{\pm0.29}$ | $4.40_{\pm0.34}$ | $4.05_{\pm0.25}$ | $2.30_{\pm0.64}$ | $\mathbf{2.69}_{\pm0.23}$ |
| PGO-BEN with BMA ($\beta = (0.5, 0.5)$) | $\mathbf{4.32}_{\pm0.13}$ | $\mathbf{4.31}_{\pm0.29}$ | $\mathbf{3.95}_{\pm0.28}$ | $\mathbf{2.24}_{\pm0.51}$ | $2.80_{\pm0.44}$ |
| $\Delta$ | -0.39 | -0.09 | -0.20 | -0.06 | +0.11 |

Table 6: **Comparison of value of $\beta$.** Superior stability-plasticity trade-off observed when $\beta = (0.5, 0.5)$. Results are mean of 3 seeds.

| Shots | $\beta = (0.3, 0.3)$ | | $\beta = (0.5, 0.5)$ | | $\beta = (0.7, 0.7)$ | | $\beta = (1, 1)$ | |
|---|---|---|---|---|---|---|---|---|
| | AA*(↑) | FA*(↑) | AA*(↑) | FA*(↑) | AA*(↑) | FA*(↑) | AA*(↑) | FA*(↑) |
| **1-shot** | 73.96 | 63.40 | **74.27** | **63.76** | 73.97 | 63.38 | 74.17 | 63.74 |
| **2-shot** | 74.32 | 63.79 | **74.36** | **63.92** | 74.29 | 63.74 | 74.32 | 63.86 |
| **4-shot** | 74.46 | 63.89 | **74.53** | **64.17** | 74.42 | 63.84 | 74.47 | 64.09 |
| **8-shot** | **74.70** | 64.18 | 74.61 | **64.28** | 74.64 | 64.05 | 74.49 | 64.00 |
| **Average** | 74.36 | 63.81 | **74.44** | **64.03** | 74.33 | 63.75 | 74.36 | 63.92 |

The choice of $\beta$ controls the temporal emphasis in the ensembling process: lower values emphasize early and late stages, while higher values favor mid-training checkpoints. Weighting curves are visualized in ***Sup. Mat.*** Empirically, $\beta = (0.5, 0.5)$ yields the best average FA across all supervision levels, highlighting that moderately bi-modal weighting ($\beta < 1$) offers better stability in BMA, validating its design choice as a robust temporal smoothing mechanism in PGO-BEN.

Table 7: **Analyzing the impact of the stability components.** We experiment on the DomainNet dataset across $1, 2, 4,$ and $8$ -shots training examples. Neglecting the BMA component results in significant forgetting of the knowledge of the previous domains. Results are mean of 3 seeds.

| Stability components | | 1-shot | | 2-shot | | 4-shot | | 8-shot | |
|---|---|---|---|---|---|---|---|---|---|
| Gradient Distill | BMA | AA*(↑) | FA*(↑) | AA*(↑) | FA*(↑) | AA*(↑) | FA*(↑) | AA*(↑) | FA*(↑) |
| ✓ | ✗ | 73.75 | 62.89 | 74.07 | 63.11 | 74.05 | 62.96 | 73.97 | 62.38 |
| ✗ | ✓ | 72.83 | 61.87 | 73.74 | 63.05 | **76.44** | 63.52 | 74.42 | 63.83 |
| ✓ | ✓ | **74.27** | **63.76** | **74.36** | **63.92** | 74.53 | **64.17** | **74.61** | **64.28** |

Table 8: **Prompting Configurations.** We compare unimodal and multimodal setups, finding that conditioning text prompts on vision tokens improves generalization under domain shift. Results are mean of 3 seeds

| Prompting | | 1-shot | | 2-shots | | 4-shots | | 8-shots | |
|---|---|---|---|---|---|---|---|---|---|
| Technique | Encoder | AA*↑ | FA*↑ | AA*↑ | FA*↑ | AA*↑ | FA*↑ | AA*↑ | FA*↑ |
| Unimodal | $\mathcal{F}_t$ | 70.34 | 59.33 | 71.11 | 59.68 | 72.41 | 61.92 | 72.96 | 62.21 |
| | $\mathcal{F}_v$ | 72.02 | 62.35 | 71.85 | 62.15 | 71.98 | 62.26 | 72.27 | 62.21 |
| Multi-modal | $\{\mathcal{F}_t, \mathcal{F}_v\}$ | 72.52 | 62.13 | 72.98 | 62.51 | 73.33 | 62.62 | 73.43 | 62.23 |
| | $\mathcal{F}_t \rightarrow \mathcal{F}_v$ | 71.50 | 61.67 | 70.99 | 61.28 | 71.60 | 61.85 | 72.50 | 62.79 |
| | $\mathcal{F}_v \rightarrow \mathcal{F}_t$ (**Ours**) | **74.27** | **63.76** | **74.36** | **63.92** | **74.53** | **64.17** | **74.61** | **64.28** |

**(d) Impact of Stability Components.** We ablate the two stability components of PGO-BEN: Proxy Guided Orthogonalization and BMA. As shown in Table 7, using Proxy Guided Orthogonalization alone may be suboptimal when $\mathcal{D}_t$ differs significantly from previous domains, leading to unreliable $G_{\text{prev}}$. BMA, based solely on $G_{\text{curr}}$, adapts better to new data and reduces overfitting through temporal smoothing. Their combination consistently yields the best performance across all supervision settings, achieving a stronger balance between adaptation and retention, especially under label scarcity.

**(e) Effectiveness of Multi-Modal Prompting.** Table 8 compares PGO-BEN's vision-conditioned prompting with uni-modal and other multi-modal baselines: Independent Prompting and Text-conditioned Vision Prompting-MaPLE*. Multimodal prompting learns better representation as compared to Unimodal

representation. MaPLE* conditions $\mathcal{F}_V$'s prompts on $\mathcal{F}_T$, thus performing sub-optimal in capturing visual domain-shift. PGO-BEN outperforms all variants across shots, with Fig. 4a showing higher similarity between learnt embeddings and better retention of knowledge, supporting our choice of conditioning for better domain invariance and generalization.

**Extended ablations**, including prompt lengths, depth of encoder synergy, and observing novel classes during inference, further analysis on BMA, are mentioned in ***Sup. Mat.***

## 5 Takeaways

We addressed the underexplored challenge of FSDIL, a setting critical for deploying continual learning systems in dynamic and low-supervision environments. We proposed PGO-BEN, a unified framework integrating multi-modal prompting, proxy-guided orthogonalization, and Beta-based ensembling. Our method demonstrates strong resilience to catastrophic forgetting while enabling efficient adaptation under severe domain shifts. Comprehensive evaluations across multiple benchmarks confirm the effectiveness and generalization of PGO-BEN, highlighting the value of stabilizing gradient trajectories, and adaptively balancing stability and plasticity. Furthermore, the framework's ability to scale across domain variations positions it is a practical solution for real-world continual learning tasks. **Future directions** include extending PGO-BEN to handle unlabeled domain adaptation scenarios, and improving sample-efficiency further via generative replay or self-supervised objectives.

**Broader Impact and Limitation:** This work can enable AI to adapt in critical data-scarce fields (e.g., healthcare, robotics), though risks of flawed adaptation or model bias are data dependent. The method assumes fixed label spaces across domains, and its long-term scalability to numerous, highly diverse domains needs further study.

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

# A Appendix

We discuss the additional details like dataset details, evaluation metrics and additional results in the appendix.

# B Contents in this Supplementary document.

This supplementary document provides detailed insights and analyses to support the main paper. The contents are organized as follows:

- **Section C: Proxy-Guided Orthogonalization (PGO)**
  Elaborates on the Proxy Guided Orthogonalization mechanism, connecting it with existing orthogonality-based continual learning literature. It also presents empirical observations that justify the chosen hyperparameters.

- **Section D: Theoretical Justification via PAC-Bayesian Framework**
  Discusses the theoretical underpinnings of our proposed method, PGO-BEN, utilizing the PAC-Bayesian framework to provide generalization guarantees.

- **Section E: Comparative Analysis of EMA and BMA**
  Offers a theoretical comparison between Exponential Moving Average (EMA) and Beta-based Moving Average (BMA). We analyze prediction variance across domain sequences and assess representation stability, especially under significant domain shifts.

- **Section F: Distinction of our approach with Multi-Task Learning**
  Discusses the fundamental differences of our approach with respect to the Multi-Task Learning from which our method takes inspiration. We highlight the aspects where our formulation differs from it, with experimental results highlighting that direct application of MTL method is sub-optimal.

- **Section G: Dataset and Domain Order Details**
  Outlines the datasets used and the specific domain sequences followed in our experiments.

- **Section H: Evaluation Metrics**
  Details the metrics employed for evaluation, providing formulas and explanations for clarity.

- **Section I: Implementation and Hardware Details**
  Discusses the implementation specifics of our method and the hardware configurations utilized during experimentation.

- **Section J: Algorithm Pseudocode**
  Presents the pseudocode of our proposed algorithm, offering a step-by-step procedural understanding.

- **Section K: Model agnostic nature of our methodology**
  Experimentally verifies that our method is model-agnostic and can be applied to any CLIP like architecture with Transformer based backbone in the encoders. Specifically, we re-run several baselines with ViTL/14 backbone of the Vision encoder of the CLIP model.

- **Section L: Comparison with Zero-shot CLIP with manual prompt**
  Presents the results obtained by prompting CLIP-ViTB/16 model with various manual prompts for all the benchmark datasets. The results highlight that existing large-scale models require careful adaptation algorithms and pre-trained weights with manual prompts provide sub-optimal performance. This also highlight that manual prompting is very hard in fine-grain datasets like CoRE50.

- **Section M: Comprehensive Results**
  Provides detailed results for 1, 2, 4, and 8-shot settings across three benchmark datasets: DomainNet, CORe50, and CDDB-Hard. Additionally, we present results corresponding to a seed value of 2 for all datasets.

- **Section O: Encoder-Synergy Module Depth Analysis**
  Examines the performance implications of varying the depth of the Encoder-Synergy module.

- **Section P: Prompt Length Modulation**
  Analyzes how changes in prompt length affect performance, providing insights into optimal configurations.

- **Section Q: Novel Class Inference**
  Explores scenarios where novel classes are introduced during inference, demonstrating that PGO-BEN effectively recognizes new classes, attributed to the robust prior knowledge from CLIP.

- **Section R: Experiments with 5 seeds**
  Explores the effect of using 5 seeds instead of 3 seeds. We rerun all the baselines for all the datasets across all the shots and report the individual score along with the average.

## C   Further Discussions on Proxy-Guided Orthogonalization

We address concerns regarding the robustness, theoretical justification, and convergence properties of our Proxy-Guided Orthogonalization (PGO) strategy, designed to enhance stability across sequential domain shifts in the Few-Shot Domain-Incremental Learning (FSDIL) setting.

A primary concern is the reliability of predictions from the previous model $\mathcal{M}_{t-1}$ when adapting to a new domain $\mathcal{D}_t$, especially when $\mathcal{D}_t$ significantly differs from prior domains $\{\mathcal{D}_1, \ldots, \mathcal{D}_{t-1}\}$. To mitigate this, PGO employs a soft directional filter rather than a hard constraint. Specifically, when the cosine angle $\psi$ between the current gradient $G_{\text{curr}}$ and the previous gradient $G_{\text{prev}}$ exceeds $90°$, indicating potential conflict, we project out the conflicting component and retain only the orthogonal component $G_{\text{curr}}^{\perp}$ (see Fig. 5). This approach ensures that adaptation to new domains does not adversely affect previously acquired knowledge, thus enhancing stability.

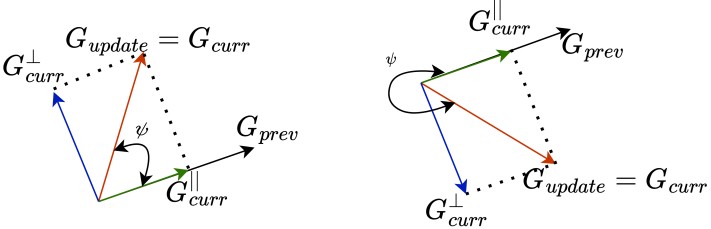

$G_{curr}^{\parallel}$ **along** the direction of previous knowledge $G_{prev}$
(doesn't interfere with previous knowledge)

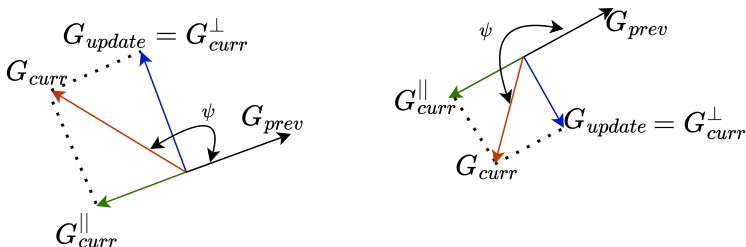

$G_{curr}^{\parallel}$ **opposite** to the direction of previous knowledge $G_{prev}$
(interfers with previous knowledge)

Figure 5: **Illustration of conflicting and non-conflicting gradient update scenarios with respect to prior knowledge.** We illustrate the scenarios where the current adaptation may or may not interfere with prior knowledge. When $\psi \in [-\frac{\pi}{2}, \frac{\pi}{2}]$, there is no conflict, allowing direct updates. In cases of conflict, updates are made in the orthogonal direction $G_{\text{curr}}^{\perp}$.

Unlike prior methods that rigidly project gradients into stored subspaces or freeze parameters Farajtabar et al. (2020); Saha et al. (2021); Liang & Li (2024; 2023), our approach allows for flexible adaptation even under significant domain shifts. Empirical results demonstrate strong retention of prior knowledge despite domain divergence.

We adopt a fixed cosine angle threshold of 90° to detect conflicting gradients. This choice is simple, interpretable, and aligns with orthogonality-based continual learning literature Farajtabar et al. (2020); Saha et al. (2021); Liang & Li (2024; 2023). To assess sensitivity, we perform ablations by varying the threshold across multiple angular deviations. The results, detailed in Table 9, indicate that PGO's performance remains robust across a range of threshold values, validating the effectiveness of our chosen threshold.

Table 9: **Using 90° as a cosine similarity threshold.** The choice of 90° as the threshold to identify conflicting knowledge among $G_{\text{prev}}$ and $G_{\text{curr}}$ is observed to empirically also enhance the stability-plasticity tradeoff.

|  | $\psi \le 45°$ & $\psi \ge 315°$ | | $\psi \le 75°$ & $\psi \ge 285°$ | | $\psi \le 90°$ & $\psi \ge 270°$ | | $\psi \le 110°$ & $\psi \ge 250°$ | |
|---|---|---|---|---|---|---|---|---|
|  | **AA\*↑** | **FA\*↑** | **AA\*↑** | **FA\*↑** | **AA\*↑** | **FA\*↑** | **AA\*↑** | **FA\*↑** |
| **1-shot** | 73.77 | 62.89 | 73.89 | 62.95 | **74.27** | **63.76** | 74.11 | 63.01 |
| **4-shot** | 74.40 | 64.02 | 74.54 | 64.09 | **74.93** | **64.48** | 74.46 | 64.01 |

Our approach operates within the CLIP framework, where the vision and text encoders remain frozen, and only lightweight prompt and adapter parameters are updated. This low-dimensional setting mitigates

optimization complexity. While formal convergence guarantees for non-convex losses are challenging, the cosine-based gating in PGO ensures that updates do not destructively interfere with previously optimized directions, leading to smoother loss trajectories. Moreover, PGO functions as a directional regularizer, avoiding gradient interference without introducing projection errors common in subspace-based continual learning.

In summary, Gradient-Aligned Distillation is a lightweight, theoretically motivated, and empirically stable mechanism for continual adaptation under domain shifts. It effectively handles domain divergence, is robust to threshold variations, and supports convergence in practice, even in non-convex prompt learning setups.

## D Theoretical Justification of PGO-BEN via PAC-Bayesian Framework

We formally justify the generalization behavior of PGO-BEN under the FSDIL setting using PAC-Bayesian analysis. The bound characterizes generalization performance for stochastic predictors trained on finite samples under a prior–posterior distributional framework.

Let $\mathcal{H}$ denote the hypothesis space parameterized by model weights $\theta$, and let $\mathcal{D}_t$ be the domain distribution at session $t$. Consider a bounded loss $\ell : \mathcal{H} \times \mathcal{X} \times \mathcal{Y} \to [0, 1]$, empirical risk $\hat{\mathcal{L}}_{\mathcal{D}_t}(h) = \frac{1}{n} \sum_{i=1}^{n} \ell(h, x_i^t, y_i^t)$, and expected risk $\mathcal{L}_{\mathcal{D}_t}(h) = \mathbb{E}_{(x,y) \sim \mathcal{D}_t}[\ell(h, x, y)]$.

**Theorem 1** (PAC-Bayes Generalization Bound McAllester (1999)). *Let $\pi$ be a prior distribution over $\mathcal{H}$, and $\rho$ be a data-dependent posterior. Then, for any $\delta \in (0, 1)$, with probability at least $1 - \delta$ over the choice of sample $S \sim \mathcal{D}_t^n$, we have:*

$$\mathbb{E}_{h \sim \rho}[\mathcal{L}_{\mathcal{D}_t}(h)] \leq \mathbb{E}_{h \sim \rho}[\hat{\mathcal{L}}_{\mathcal{D}_t}(h)] + \sqrt{\frac{KL(\rho\|\pi) + \log \frac{2\sqrt{n}}{\delta}}{2n}}.$$

**Prior $\pi$:** We define the prior $\pi$ as the parameter distribution of the pretrained CLIP model ($\theta_0$) before adaptation begins. Due to CLIP's exposure to broad domain diversity, this prior is semantically rich and well-aligned with the hypothesis space for downstream domains, particularly useful in the few-shot regime where $\rho$ must stay close to $\pi$.

**Posterior $\rho$ via BMA:** The posterior $\rho$ is implicitly constructed as a mixture of model checkpoints across training steps:

$$\rho = \sum_{t'=0}^{T'} \alpha_{t'} \, \delta_{\mathcal{M}_{t'}}, \quad \text{where} \quad \alpha_{t'} \propto \text{Beta}(\beta, \beta) \left( \frac{t'+0.5}{T'+1} \right).$$

This formulation yields a smoothed, trajectory-aware posterior that reduces overfitting to the final iterate and aligns with posterior averaging methods shown to tighten PAC-Bayesian bounds Dziugaite & Roy (2017); Wu et al. (2019).

**Stability via Proxy-Guided Orthogonalization:** The cosine-based masking of gradient directions in PGO prevents catastrophic drift from $\mathcal{M}_{t-1}$, enforcing update stability without requiring stored gradients. This aligns with algorithmic stability theory Bousquet & Elisseeff (2002), which bounds the empirical–expected risk gap and improves generalization.

Summarily, each component of PGO-BEN contributes to a tighter PAC-Bayesian bound:

- A strong prior $\pi$ via CLIP reduces the complexity term $KL(\rho\|\pi)$;

- The BMA-based posterior $\rho$ smooths model updates, lowering variance in empirical loss;

- Gradient-aligned updates stabilize training, narrowing the empirical–expected loss gap.

While deriving exact closed-form bounds in deep networks remains intractable Bousquet & Elisseeff (2002); Dziugaite & Roy (2017), our formulation aligns with established PAC-Bayes theory and is supported by robust empirical generalization across non-i.i.d. domain streams.

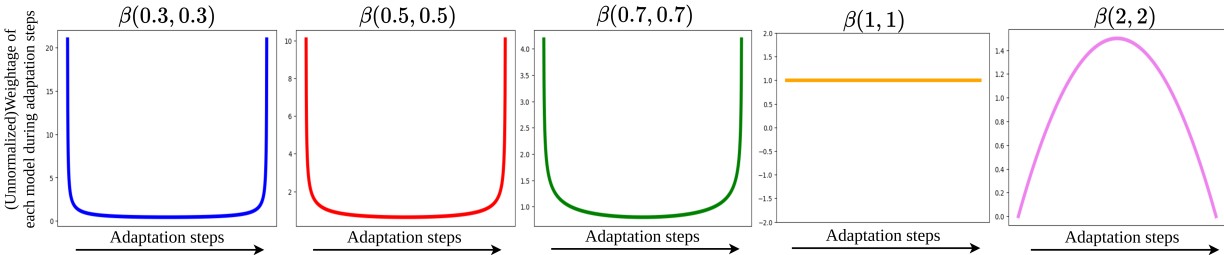

Figure 6: **Beta-distribution with different parameters.** Variation of the Beta-distribution curve with different parameters.

# E   Theoretical and Empirical Analysis of Beta-based Moving Average (BMA)

EMA is a first-order recursive smoother of model parameters:

$$\theta_t^{\text{EMA}} = \lambda\theta_t + (1-\lambda)\theta_{t-1}^{\text{EMA}},$$

where $\lambda \in [0,1]$ is the decay parameter. Its recursive nature gives exponentially diminishing weights to early iterates, rapidly discarding useful information from past domains.

BMA, in contrast, uses a non-recursive weighted sum of intermediate model checkpoints:

$$\theta^{\text{BMA}} = \sum_{t'=0}^{T'} \alpha_{t'}\theta_{t'}, \quad \alpha_{t'} \propto \text{Beta}(\beta, \beta)\left(\frac{t'+0.5}{T'+1}\right).$$

This induces a *non-monotonic, symmetric weighting* over time, explicitly preserving early-stage information while still incorporating late-stage adaptation. Unlike Gaussian posterior smoothing or Bayesian ensembling, which require approximate posterior distributions over parameters (e.g., Laplace or variational), BMA is an *implicit posterior smoother* operating over deterministic iterates. This avoids costly uncertainty estimation or sampling, while still capturing temporal uncertainty through weighting. This choice is deliberate: in FSDIL, labeled data is too sparse to fit full Bayesian posteriors per domain, and BMA provides an efficient surrogate.

Figure 6 describes the shape of the Beta-distribution curve. As we see, with $\beta < 1$, the models at either end of the adaptation strategy are given more weightage. This aligns with the intuition that model state during the initial iterations of the adaptation step in domain $\mathcal{D}_t$ is likely to have more knowledge about domains $\{\mathcal{D}_1, \cdots, \mathcal{D}_{t-1}\}$, and hence should be given more weightage to ensure that even after adaptation, the model $\mathcal{M}_t$, adapted on domain $\mathcal{D}_t$ preserves the knowledge of domains $\{\mathcal{D}_1, \cdots, \mathcal{D}_{t-1}\}$. $\beta = 1$, gives equal weightage to all the intermediate states.

**Variance Reduction:**  Let $\bar{\theta} = \mathbb{E}_{t' \sim \alpha}[\theta_{t'}]$. The total variance under BMA is:

$$\text{Var}_\alpha(\theta_{t'}) = \mathbb{E}_\alpha[\|\theta_{t'} - \bar{\theta}\|^2],$$

which, for symmetric $\beta < 1$, gives more uniform support across the training trajectory, reducing the bias toward terminal points that plagues EMA. This stabilization is critical in FSDIL, where prior domain knowledge must not be erased. In Fig. **??** (main paper), we show the prediction variance comparisons between BMA and EMA on the prediction on the test set of Real domain dataset as the model keeps adapting to a sequence of domains, in 4-shot scenario. BMA is found to reduce the prediction variance more effectively, thus maintaining steady performance as compared to EMA much better. We also compute the change in cosine-similarity of the output of text and vision encoder, as the model has to adapt to a large domain-shift. After learning about the Real domain, the average cosine-similarity on the Real domain test set is 0.3480. As we observe, for both the adaptation scenarios (Real $\rightarrow$ Clipart and Real $\rightarrow$ Sketch), the drop in the cosine similarity of the text and vision encoder representations for the EMA model is higher compared to BMA, which indicates that the representations learnt by EMA is more stable than EMA.

|        | Ours      | MTL   |
|--------|-----------|-------|
| 1-shot | **62.33** | 55.07 |
| 2-shot | **62.42** | 55.16 |
| 8-shot | **64.54** | 56.63 |

Table 10: Comparison of our method and MTL Yu et al. (2020) on DomainNet dataset, with respect to $AA_T$. We observe that our method is significantly better than the applying MTL approach directly, which results in more conflicting graident updates, hampering the learning.

The choice of $\beta = (0.5, 0.5)$ corresponds to the arcsine distribution, which maximally weights both early and late checkpoints:

$$\text{Beta}(0.5, 0.5) \sim \frac{1}{\pi\sqrt{x(1-x)}}, \quad x \in (0, 1).$$

This is particularly suitable in continual learning where: - *Early iterates capture prior domain knowledge*, and - *Later iterates specialize to the current domain.*

In summary, BMA offers a principled, interpretable, and efficient strategy to smooth adaptation across domains in FSDIL. It reduces variance, preserves early domain knowledge, and improves generalization stability over EMA. While formal convergence bounds remain an open direction, our empirical and intuitive justification strongly supports its use over classical EMA or probabilistic smoothing methods in the continual few-shot regime.

## F    Distinction with Multi-Task Learning

While our method is inspired by Yu et al. (2020), it significantly departs from standard multi-task learning (MTL) due to the unique constraints of FSDIL.

In Multi-Task Learning Yu et al. (2020), all task data is available simultaneously, allowing per-task gradient computation and direct conflict resolution. In contrast, FSDIL restricts access to only the current domain $\mathcal{D}_t$, with no replay or task labels, and faces few-shot supervision and unconstrained domain shifts—conditions where direct gradient projection (as in Liang & Li (2024)) becomes unreliable.

Our key innovations are as follows:

- We use the frozen model $\mathcal{M}_{t-1}$ as a proxy for prior domain knowledge. By passing current inputs $\mathcal{D}_t$ through $\mathcal{M}_{t-1}$, we approximate prior gradients without accessing old data.

- We compare gradients from $\mathcal{M}_t$ (CE loss) and $\mathcal{M}_{t-1}$ (KL loss). When conflicting directions are detected, we project the current CE loss gradient orthogonally to preserve prior knowledge—achieving forgetting mitigation without memory or subspace estimation.

- Since $\mathcal{M}_{t-1}$ is evaluated on an unseen domain, its gradients may be noisy. To stabilize updates, we introduce BMA, which adaptively ensembles model states and improves retention under shift.

Thus, while inspired by MTL conflict resolution, our formulation is fundamentally adapted to FSDIL: exemplar-free, domain-incremental, and few-shot. We will further clarify this in the revised version.

We conducted an experiment where data from all domains was introduced jointly and trained using Yu et al. (2020), treating each domain as a separate task—referred to as MTL*. We compared this against our method, which observes each domain sequentially. Average accuracy (AA) across all domains is reported below.

## G    Dataset details

We perform our experiments on three standard Domain Incremental Learning(DIL) benchmarks. The detailed descriptions and statistics These datasets are as follows:

- **CDDB** Li et al. (2023) is a dataset used for continuous deepfake detection, where the DIL objective involves recognizing authentic and fake images across different domains. We adopted the Hard Setting from Wang et al. (2022a), requiring learning on 5 continuous deepfake detection domains: GauGAN, BigGAN, WildDeepfake, WhichFaceReal, and SAN. This entails approximately 27,000 images. The domain order followed aligns with Wang et al. (2022a), i.e. GauGAN → BigGAN → WildDeepfake → WhichFaceReal → SAN.

- **CORe50** Lomonaco & Maltoni (2017) is designed for continuous object recognition, consisting of 11 domains, each with 50 categories. In DIL, we perform incremental learning on the first eight domains, as s1 → s2 → s3 → s4 → s5 → s6 → s7 → s8.

- **DomainNet** Peng et al. (2019) is a domain adaptation dataset commonly used as a benchmark for DIL methods. It comprises 6 domains, each with 345 categories. The domain order is the same as Rakshit et al. (2022), followed by Real → Painting → Clipart → Sketch → Quickdraw → Infograph, which follows an incrementally more difficult domain to learn.

## H  Evaluation metric details

After adapting to a domain $\mathcal{D}_t$ we evaluate the performance on domains $\{\mathcal{D}_1 \cdots \mathcal{D}_t\}$. To measure the effectiveness of our method towards handling the stability-plasticity trade-off, we use two standard metrics, Average Accuracy (AA) and Forgetting Alleviation (FA). Average Accuracy for domain $\mathcal{D}_t$ is defined as

$$AA_t = \sum_{i=1}^{t} A_{i,t} \tag{7}$$

where $A_{i,t}$ denotes the accuracy obtained by the model on the $i$-th domain adapting to $t$-th domain.

We used $AA^*$ as our metric which is defined as

$$AA^* = \frac{1}{\mathcal{N}} \sum_{t=1}^{\mathcal{N}} AA_t \tag{8}$$

where $N$ denotes the number of domains. This metric provides a more comprehensive measure of how the performance varies across all the training sessions and thus reduces the bias of only checking the performance on the last session. As $AA^*$ is an average of average values, a little improvement indicates much superior performance than the other counterpart.

Forgetting Alleviation for domain $\mathcal{D}_t$ is defined as

$$FA_t = \sum_{i=t+1}^{\mathcal{N}} A_{t,i} \tag{9}$$

This measures the average performance of the model on the domain $\mathcal{D}_t$ after being adapted to subsequent domains $\mathcal{D}_{t+1} \cdots \mathcal{D}_{\mathcal{N}}$. We used $FA^*$ as our metric which is defined as

$$FA^* = \frac{1}{\mathcal{N}} \sum_{t=1}^{\mathcal{N}} FA_t \tag{10}$$

We present a walthrough of the calculation of the metrics in Fig 7, taking a toy example of three domains $\{\mathcal{D}_1, \mathcal{D}_2, \mathcal{D}_3\}$, with the domain sequence being $\{\mathcal{D}_1 \rightarrow \mathcal{D}_2 \rightarrow \mathcal{D}_3\}$.

## I  Implementation details

We maintained CLIPRadford et al. (2021) ViT-B/16 as our backbone architecture for all the datasets, and all baselines (except in Sec. K). We used SGD as our optimizer with an initial learning rate of 0.002. The input images are resized to $(224 \times 224)$ for all the baselines and our proposed method. We did our training on a single NVIDIA RTX A6000 48GB-GPU, and used Pytorch as our Deep learning framework, running the models for 20 epochs.

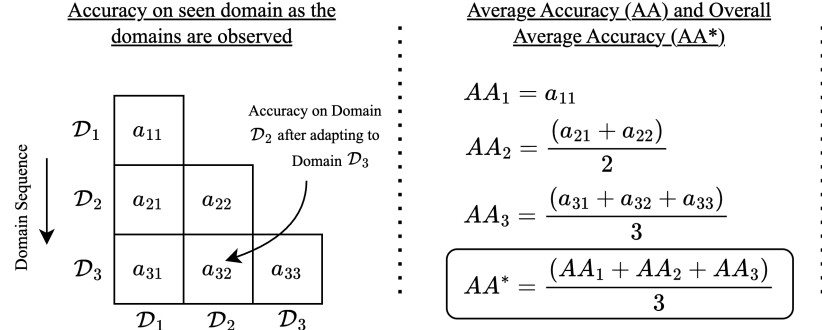

Figure 7: **Metric calculation walkthrough:** A simple walkthrough using an example of three domains. We detail the individual steps to calculate $AA^*$ and $FA^*$.

## J    Algorithm

We detail our learning process in the first session and in the incremental sessions with few-shot labeled examples in the form of a pseudo-code in Algorithm 1.

---

**Algorithm 1** Training and inference procedure of PGO-BEn

---

**Require:** Dataset $\{D_1, D_2, ..., D_\mathcal{N}\}$, Model $\mathcal{M}$, max epochs *max_epoch*, parameter $\beta$ for Beta distribution.

1: **for** $t = 1$ to $\mathcal{N}$ **do**
2:      $(x^t, y^t) \leftarrow D_t$
3:      **if** $t == 1$ **then**                                               ▷ *First domain*
4:          **for** $epoch = 1$ to *max_epoch* **do**
5:              **for** $j = 1$ to $|D_1|$ **do**
6:                  $y_{pred} \leftarrow \mathcal{M}_1(x_j^1)$
7:                  $\mathcal{L}_{ce} \leftarrow \text{crossentropy}(y_{pred}, y_j^1)$
8:                  $\mathcal{L}_{ce}.\text{backward}()$
9:                  optimizer.step()                        ▷ *Updates all the learnable parameters.*
10:              **end for**
11:          **end for**
12:          Use $\mathcal{M}_1$ for inference at $t = 1$.
13:      **else**                                       ▷ *Incremental domains with few-shot samples per class.*
14:          $\mathcal{M}_t \leftarrow \mathcal{M}_{t-1}$      ▷ *Initializing model to be adapted to domain $\mathcal{D}_t$ using the model which has been adapted to $\{\mathcal{D}_1 \cdots \mathcal{D}_{t-1}\}$*
15:          $\mathcal{M}_t^{BMA} \leftarrow \mathcal{M}_{t-1}$          ▷ *Initializing the BMA model using the model which has been adapted to $\{\mathcal{D}_1 \cdots \mathcal{D}_{t-1}\}$*
16:          **for** epoch $= 1$ to max_epoch **do**
17:              $iter \leftarrow 0$
18:              **for** $j = 1$ to $|D_t|$ **do**
19:                  $y_{pred}^t \leftarrow \mathcal{M}_t(x_j^t)$ ▷ *Prediction probability vector from the model we are adapting to domain $\mathcal{D}_t$*
20:                  $y_{pred}^{t-1} \leftarrow \mathcal{M}_{t-1}(x_j^t)$      ▷ *Prediction probability vector from the frozen model which has been adapted sequentially $\mathcal{D}_1 \cdots \mathcal{D}_{t-1}$*
21:                  $\mathcal{L}_{ce} \leftarrow \text{CROSSENTROPY}(y_{pred}^t, y_j^t)$
22:                  $\mathcal{L}_{kd} \leftarrow \text{KL - Divergence}(y_{pred}^t, y_{pred}^{t-1})$
23:                  $G_{curr} \leftarrow$ Gradient of $\mathcal{L}_{ce}$
24:                  $G_{prev} \leftarrow$ Gradient of $\mathcal{L}_{kl}$
25:                  Compute $\psi$, angle between $G_{curr}$ & $G_{prev}$ for each learnable paramters of $\mathcal{M}_t$.
26:                  **if** $\psi < 90°$ or $\psi > 270°$ **then**
27:                      $G_{update} \leftarrow G_{curr}$                  ▷ *No conflict with knowledge of previous domains.*
28:                  **else**
29:                      Decompose $G_{curr}$ into $G_{curr}^{||}$ and $G_{curr}^{\perp}$ which denote component of $G_{curr}$ parallel to $G_{prev}$ and perpendicular to $G_{prev}$ respectively.
30:                      $G_{update} \leftarrow G_{curr}^{\perp}$
31:                  **end if**
32:                  optimizer.step()
33:                  Obtain $\alpha_{t'}$ from Equation 9 with $t' = iter$.
34:                  Compute $\gamma_t \leftarrow \frac{\alpha_{t'}}{\sum_{k=0}^{t'} \alpha_k}$
35:                  Compute $\mathcal{M}_{t'}^{BMA} \leftarrow (1 - \gamma)\mathcal{M}_{t'-1}^{BMA} + \gamma \cdot \mathcal{M}_{t'}$
36:                  $iter \leftarrow iter + 1$
37:              **end for**
38:          **end for**
39:          Use $\mathcal{M}_t^{BMA}$ for inferencing on domains $\{\mathcal{D}_1 \cdots \mathcal{D}_t\}$ seen so far.
40:      **end if**
41: **end for**

---

# K Other backbone models (CLIP ViT-L)

To asses the model-agnostic effectiveness of our proposed method, we conducted additional experiments using the CLIP ViT-L/14 backbone and compared with one regularizer based baseline and gradient approximation technique respectively. We evaluate our method across various levels of supervision, specifically in 1-shot, 2-shot and 4-shot settings, in Table 11. We observed that our method consistently outperforms the baseline methods across all supervision levels.

Table 11: **Performance with CLIP ViT-L/14 backbone.** We replace the backbone of PGO-BEn and baseline methods with CLIP ViT-L/14 to assess generality. PGO-BEn maintains the best performance under all supervision levels, validating its backbone-agnostic continual adaptation capability.

| Method | 1-shot | | 2-shots | | 4-shots | |
|---|---|---|---|---|---|---|
| | AA*↑ | FA*↑ | AA*↑ | FA*↑ | AA*↑ | FA*↑ |
| EwC | 76.07 | 65.71 | 75.98 | 65.24 | 75.78 | 64.43 |
| LwF | 76.59 | 67.45 | 77.07 | 67.33 | 76.50 | 65.87 |
| InfLORA | 76.81 | 67.50 | 77.34 | 67.72 | 76.95 | 66.57 |
| Ours | **78.12** | **68.76** | **78.15** | **68.81** | **78.25** | **68.75** |

# L Comparison with Zero-shot CLIP with various manual prompt

In Table 12, 13 and 14 we discuss the performance obtained upon changing the manual prompt. As we observe, the performance varies quite drastically for all the benchmarks. This highlights that, large-scale pretrained models like CLIP fail to adapt to changing domains, or even fine-grain classification like identifying identity of individual objects. It is thus required to design efficient methods to train the pre-trained models to adapt to this evolving domain scenarios.

Table 12: DomainNet

| Prompt | Real | Painting | Clipart | Sketch | Quickdraw | Infograph |
|---|---|---|---|---|---|---|
| a photo of a — | 83.19 | 63.00 | 69.86 | 64.12 | 13.91 | 49.08 |
| Ours (1-shot) | 86.37 | 68.77 | 75.68 | 67.87 | 20.84 | 54.47 |

Table 13: CDDB-Hard

| Prompt | GauGAN | BigGAN | Wild | WhichfaceisReal | SAN |
|---|---|---|---|---|---|
| a photo of a — image | 57.75 | 52.75 | 52.01 | 68.50 | 50.60 |
| a — image | 57.00 | 53.75 | 51.53 | 68.50 | 49.40 |
| Ours (4-shot) | 90.45 | 86.12 | 56.51 | 70.50 | 61.44 |

Table 14: CoRE50

| Prompt | s1 | s2 | s3 | s4 | s5 | s6 | s7 | s8 |
|---|---|---|---|---|---|---|---|---|
| a photo of a — | 11.67 | 12.20 | 9.20 | 11.33 | 10.53 | 8.20 | 10.20 | 9.83 |
| there is a — in this image | 14.43 | 13.73 | 11.97 | 11.37 | 12.77 | 10.50 | 12.90 | 13.73 |
| this is an image of — | 13.33 | 15.47 | 11.80 | 11.33 | 11.10 | 9.27 | 12.10 | 11.37 |
| Ours (1-shot) | 88.67 | 77.23 | 75.26 | 80.33 | 76.40 | 68.53 | 73.30 | 83.96 |

# M Result

In this section we expand out the results on DomainNet, CDDB-Hard and CoRE50 dataset that we have shown in Table 2 of the main paper. We report the average AA* and FA* across 1, 2, 4, and 8 shots in the main paper. Here we detail them individually.

Table 15 details results on DomainNet. Table 16 details obtained results on CDDB-Hard and Table 17 details obtained results on CoRE50 dataset.

The results, averaged over three seeds, are in the next page owing to the orientation.

Table 15: **Comparison with existing DIL benchmarks on DomainNet dataset across** 1, 2, 4 **and** 8-**shots.** We report the **AA\*** and **FA\*** values for comparison, which indicate 'Overall Average Accuracy' and 'Overall Forgetting Alleviation.' The highest performance is shown in **bold**, with the second highest underlined. PGO-BEn performs superior compared to all the baselines across the varying levels of supervision, highlighting the effectiveness of the proposed methodology.* indicates reimplemented with CLIP.

| Method | Prompt pool | Backbone | 1-shot | | 2-shot | | 4-shot | | 8-shot | |
|---|---|---|---|---|---|---|---|---|---|---|
| | | | AA*(↑) | FA*(↑) | AA*(↑) | FA*(↑) | AA*(↑) | FA*(↑) | AA*(↑) | FA*(↑) |
| DyToxDouillard et al. (2021) | × | ViT | $29.94_{\pm0.68}$ | $18.72_{\pm0.59}$ | $29.20_{\pm0.86}$ | $18.20_{\pm0.66}$ | $35.59_{\pm1.1}$ | $22.57_{\pm0.62}$ | $29.71_{\pm0.96}$ | $18.58_{\pm0.68}$ |
| LwF*Li & Hoiem (2017) | × | CLIP | $72.13_{\pm0.55}$ | $61.51_{\pm1.20}$ | $72.26_{\pm0.75}$ | $61.26_{\pm0.67}$ | $72.08_{\pm0.68}$ | $60.47_{\pm1.21}$ | $71.77_{\pm1.31}$ | $59.58_{\pm0.65}$ |
| EwC*Kirkpatrick et al. (2017) | × | " | $71.71_{\pm1.19}$ | $60.87_{\pm1.01}$ | $70.99_{\pm0.63}$ | $59.04_{\pm0.85}$ | $70.43_{\pm0.78}$ | $57.85_{\pm1.23}$ | $70.57_{\pm1.02}$ | $57.67_{\pm0.47}$ |
| L2P* Wang et al. (2022c) | ✓ | " | $65.58_{\pm0.59}$ | $53.10_{\pm0.34}$ | $67.18_{\pm0.78}$ | $54.92_{\pm0.92}$ | $67.44_{\pm0.62}$ | $54.82_{\pm0.38}$ | $68.14_{\pm0.55}$ | $55.61_{\pm0.29}$ |
| DualPrompt* Wang et al. (2022b) | ✓ | " | $\underline{72.50}_{\pm0.56}$ | $62.25_{\pm0.74}$ | $73.10_{\pm0.85}$ | $63.18_{\pm0.67}$ | $73.81_{\pm0.77}$ | $64.00_{\pm0.68}$ | $74.44_{\pm0.46}$ | $64.58_{\pm0.94}$ |
| S-Prompt Wang et al. (2022a) | ✓ | " | $62.28_{\pm0.32}$ | $50.18_{\pm0.36}$ | $67.52_{\pm0.58}$ | $55.81_{\pm0.46}$ | $69.85_{\pm0.21}$ | $58.60_{\pm0.19}$ | $70.92_{\pm0.37}$ | $59.95_{\pm0.28}$ |
| CODA-Prompt Smith et al. (2023) | ✓ | " | $72.43_{\pm0.95}$ | $62.38_{\pm0.87}$ | $\underline{73.22}_{\pm0.62}$ | $63.10_{\pm0.76}$ | $\underline{73.87}_{\pm0.67}$ | $63.66_{\pm0.59}$ | $\underline{74.47}_{\pm0.97}$ | $64.24_{\pm0.27}$ |
| InfLORA*Liang & Li (2024) | × | " | $72.39_{\pm0.49}$ | $61.79_{\pm0.67}$ | $72.05_{\pm0.86}$ | $60.90_{\pm0.77}$ | $71.83_{\pm0.64}$ | $60.04_{\pm0.95}$ | $71.47_{\pm0.37}$ | $58.93_{\pm0.65}$ |
| CP-PromptFeng et al. (2024) | ✓ | " | $70.02_{\pm0.61}$ | $58.16_{\pm0.58}$ | $71.58_{\pm0.97}$ | $60.27_{\pm0.67}$ | $72.52_{\pm0.94}$ | $61.82_{\pm0.88}$ | | |
| **PGO-BEn** | × | CLIP | $\mathbf{74.27}_{\pm0.11}$ | $\mathbf{63.76}_{\pm0.19}$ | $\mathbf{74.36}_{\pm0.25}$ | $\mathbf{63.92}_{\pm0.28}$ | $\mathbf{74.93}_{\pm0.16}$ | $\mathbf{64.48}_{\pm0.29}$ | $\mathbf{75.51}_{\pm0.17}$ | $\mathbf{66.93}_{\pm0.26}$ |
| | | Δ | +1.77 | +1.51 | +1.26 | +0.74 | +1.12 | +0.48 | +1.07 | +2.35 |

Table 16: **Comparison with existing DIL benchmarks on CDDB-Hard dataset across** 1, 2, 4 **and 8-shots.** We report the **AA\*** and **FA\*** values for comparison, which indicate 'Overall Average Accuracy' and 'Overall Forgetting Alleviation.' The highest performance is shown in **bold**, with the second highest underlined. PGO-BEN performs superior compared to all the baselines across the varying levels of supervision, highlighting the effectiveness of the proposed methodology.* indicates reimplemented with CLIP.

| Method | Prompt pool | Backbone | 1-shot | | 2-shot | | 4-shot | | 8-shot | |
|---|---|---|---|---|---|---|---|---|---|---|
| | | | AA*(↑) | FA*(↑) | AA*(↑) | FA*(↑) | AA*(↑) | FA*(↑) | AA*(↑) | FA*(↑) |
| DyToxDouillard et al. (2021) | × | ViT | $58.79_{\pm0.48}$ | $55.19_{\pm0.96}$ | $57.91_{\pm1.21}$ | $53.34_{\pm0.67}$ | $55.82_{\pm0.26}$ | $51.59_{\pm0.84}$ | $56.18_{\pm0.44}$ | $52.56_{\pm0.80}$ |
| LwF*Li & Hoiem (2017) | × | CLIP | $62.71_{\pm0.95}$ | $53.88_{\pm0.56}$ | $67.82_{\pm0.88}$ | $59.24_{\pm0.98}$ | $71.31_{\pm0.46}$ | $61.26_{\pm0.87}$ | $71.16_{\pm0.67}$ | $61.58_{\pm0.58}$ |
| EwC*Kirkpatrick et al. (2017) | × | " | $63.69_{\pm0.69}$ | $53.13_{\pm0.38}$ | $65.32_{\pm1.22}$ | $55.99_{\pm0.98}$ | $\underline{78.10}_{\pm0.87}$ | $\underline{70.63}_{\pm0.79}$ | $78.09_{\pm0.48}$ | $69.09_{\pm0.92}$ |
| L2P* Wang et al. (2022c) | ✓ | " | $64.54_{\pm0.68}$ | $58.60_{\pm0.97}$ | $\underline{74.84}_{\pm1.29}$ | $\underline{68.14}_{\pm0.69}$ | $73.24_{\pm0.87}$ | $65.52_{\pm0.89}$ | $73.27_{\pm0.93}$ | $65.43_{\pm0.79}$ |
| DualPrompt* Wang et al. (2022b) | ✓ | " | $\underline{72.12}_{\pm0.59}$ | $\underline{65.32}_{\pm0.67}$ | $72.47_{\pm0.44}$ | $65.98_{\pm0.68}$ | $73.75_{\pm0.88}$ | $67.44_{\pm0.96}$ | $\underline{75.15}_{\pm0.73}$ | $67.33_{\pm0.89}$ |
| S-Prompt Wang et al. (2022a) | ✓ | " | $63.68_{\pm0.44}$ | $58.23_{\pm0.37}$ | $64.23_{\pm0.68}$ | $59.86_{\pm0.57}$ | $65.59_{\pm0.83}$ | $61.20_{\pm0.86}$ | $67.74_{\pm0.28}$ | $61.59_{\pm0.29}$ |
| CODA-Prompt Smith et al. (2023) | ✓ | " | $71.24_{\pm0.46}$ | $60.80_{\pm0.67}$ | $71.23_{\pm0.36}$ | $60.87_{\pm0.37}$ | $70.33_{\pm0.57}$ | $60.79_{\pm0.28}$ | $69.34_{\pm0.60}$ | $59.35_{\pm0.57}$ |
| InfLORA*Liang & Li (2024) | × | " | $62.69_{\pm0.29}$ | $52.21_{\pm0.83}$ | $61.58_{\pm0.65}$ | $55.05_{\pm0.67}$ | $68.66_{\pm0.47}$ | $58.00_{\pm0.66}$ | $73.70_{\pm0.49}$ | $61.34_{\pm0.93}$ |
| CP-PromptFeng et al. (2024) | ✓ | " | $66.87_{\pm0.61}$ | $61.93_{\pm0.36}$ | $66.94_{\pm0.45}$ | $62.95_{\pm0.25}$ | $66.73_{\pm0.11}$ | $61.48_{\pm0.17}$ | $67.26_{\pm0.27}$ | $62.04_{\pm0.11}$ |
| PGO-BEn | × | CLIP | $\textbf{74.68}_{\pm0.27}$ | $\textbf{66.34}_{\pm0.14}$ | $\textbf{77.78}_{\pm0.29}$ | $\textbf{71.84}_{\pm0.21}$ | $\textbf{83.69}_{\pm0.39}$ | $\textbf{76.79}_{\pm0.10}$ | $\textbf{84.22}_{\pm0.21}$ | $\textbf{77.71}_{\pm0.19}$ |
| | | Δ | +2.56 | +1.02 | +2.94 | +3.70 | +5.59 | +6.16 | +9.07 | +8.62 |

Table 17: **Comparison with existing DIL benchmarks on CoRE50 dataset across 1, 2, 4 and 8-shots.** We report the **AA\*** and **FA\*** values for comparison, which indicate 'Overall Average Accuracy' and 'Overall Forgetting Alleviation.' The highest performance is shown in **bold**, with the second highest underlined. PGO-BEN performs superior compared to all the baselines across the varying levels of supervision, highlighting the effectiveness of the proposed methodology.* indicates reimplemented with CLIP.

| Method | Prompt pool | Backbone | 1-shot | | 2-shot | | 4-shot | | 8-shot | |
|---|---|---|---|---|---|---|---|---|---|---|
| | | | AA\*($\uparrow$) | FA\*($\uparrow$) | AA\*($\uparrow$) | FA\*($\uparrow$) | AA\*($\uparrow$) | FA\*($\uparrow$) | AA\*($\uparrow$) | FA\*($\uparrow$) |
| DyTox Douillard et al. (2021) | $\times$ | ViT | $45.06_{\pm0.46}$ | $26.94_{\pm1.24}$ | $47.75_{\pm1.33}$ | $29.83_{\pm1.13}$ | $45.14_{\pm0.87}$ | $27.30_{\pm0.95}$ | $48.33_{\pm0.64}$ | $30.83_{\pm0.48}$ |
| LwF*Li & Hoiem (2017) | $\times$ | CLIP | $59.40_{\pm0.67}$ | $52.99_{\pm0.29}$ | $66.03_{\pm0.58}$ | $57.48_{\pm0.58}$ | $64.61_{\pm0.84}$ | $57.29_{\pm0.89}$ | $67.62_{\pm0.91}$ | $63.24_{\pm0.63}$ |
| EwC*Kirkpatrick et al. (2017) | $\times$ | " | $58.43_{\pm0.83}$ | $51.89_{\pm0.59}$ | $66.47_{\pm0.30}$ | $58.11_{\pm0.41}$ | $63.55_{\pm0.73}$ | $54.80_{\pm0.27}$ | $65.22_{\pm0.86}$ | $57.62_{\pm0.47}$ |
| L2P* Wang et al. (2022c) | $\checkmark$ | " | $\underline{80.19}_{\pm0.45}$ | $\underline{77.85}_{\pm0.85}$ | $80.94_{\pm0.96}$ | $\underline{79.55}_{\pm0.73}$ | $79.42_{\pm0.63}$ | $77.98_{\pm1.33}$ | $79.00_{\pm0.98}$ | $78.06_{\pm0.78}$ |
| DualPrompt* Wang et al. (2022b) | $\checkmark$ | " | $43.83_{\pm0.48}$ | $36.35_{\pm0.84}$ | $59.53_{\pm0.45}$ | $55.28_{\pm0.98}$ | $58.37_{\pm0.25}$ | $52.39_{\pm0.24}$ | $60.73_{\pm0.84}$ | $58.16_{\pm0.77}$ |
| S-Prompt Wang et al. (2022a) | $\checkmark$ | " | $77.63_{\pm0.76}$ | $74.26_{\pm0.57}$ | $79.53_{\pm0.84}$ | $74.95_{\pm0.28}$ | $79.63_{\pm0.69}$ | $77.26_{\pm0.58}$ | $80.13_{\pm0.78}$ | $78.77_{\pm0.67}$ |
| CODA-Prompt Smith et al. (2023) | $\checkmark$ | " | $53.79_{\pm0.49}$ | $40.77_{\pm0.87}$ | $55.09_{\pm0.69}$ | $40.82_{\pm0.39}$ | $57.87_{\pm0.73}$ | $44.95_{\pm0.87}$ | $59.97_{\pm0.92}$ | $47.89_{\pm0.94}$ |
| InfLORA*Liang & Li (2024) | $\times$ | " | $57.44_{\pm0.97}$ | $50.36_{\pm0.77}$ | $66.47_{\pm1.07}$ | $57.69_{\pm0.88}$ | $63.90_{\pm0.59}$ | $58.27_{\pm0.47}$ | $73.12_{\pm0.97}$ | $66.10_{\pm0.77}$ |
| CP-PromptFeng et al. (2024) | $\checkmark$ | " | $78.28_{\pm0.49}$ | $77.23_{\pm0.36}$ | $\underline{81.65}_{\pm0.23}$ | $78.68_{\pm0.56}$ | $\underline{82.27}_{\pm1.17}$ | $\underline{81.26}_{\pm1.02}$ | $\underline{84.08}_{\pm0.67}$ | $\underline{82.79}_{\pm0.23}$ |
| **Ours** | $\times$ | CLIP | $\mathbf{83.19}_{\pm0.20}$ | $\mathbf{78.14}_{\pm0.34}$ | $\mathbf{86.73}_{\pm0.33}$ | $\mathbf{82.81}_{\pm0.29}$ | $\mathbf{87.38}_{\pm0.51}$ | $\mathbf{83.81}_{\pm0.86}$ | $\mathbf{88.43}_{\pm0.37}$ | $\mathbf{85.34}_{\pm0.66}$ |
| $\Delta$ | | | +3.00 | +0.29 | +5.08 | +3.26 | +5.11 | +2.55 | +4.35 | +2.55 |

## N Detailed results for all datasets

We detail the results of the performance of PGO-BEɴ on DomainNet Peng et al. (2019) in Table 18, CDDB-Hard Li et al. (2023) in Table 19, and about the CoRE50 dataset Lomonaco & Maltoni (2017) in Table 20, across $1, 2, 4, 8$ and $16$ shots.

Table 18: **Performance change across varying levels of supervision for DomainNet** dataset with seed = 2.

(a) 1-shot

|           | Real  | Painting | Clipart | Sketch | Quickdraw | Infograph |
|-----------|-------|----------|---------|--------|-----------|-----------|
| Real      | 88.52 | -        | -       | -      | -         | -         |
| Painting  | 87.89 | 69.52    | -       | -      | -         | -         |
| Clipart   | 87.57 | 69.46    | 75.15   | -      | -         | -         |
| Sketch    | 87.28 | 69.78    | 75.86   | 67.43  | -         | -         |
| Quickdraw | 85.45 | 67.66    | 75.62   | 66.34  | 21.09     | -         |
| Infograph | 86.37 | 68.77    | 75.68   | 67.87  | 20.84     | 54.47     |

(b) 2-shot

|           | Real  | Painting | Clipart | Sketch | Quickdraw | Infograph |
|-----------|-------|----------|---------|--------|-----------|-----------|
| Real      | 88.52 | -        | -       | -      | -         | -         |
| Painting  | 87.98 | 70.16    | -       | -      | -         | -         |
| Clipart   | 87.58 | 69.91    | 75.86   | -      | -         | -         |
| Sketch    | 87.59 | 69.83    | 75.87   | 67.55  | -         | -         |
| Quickdraw | 85.70 | 68.28    | 75.29   | 67.59  | 20.82     | -         |
| Infograph | 86.64 | 69.64    | 75.48   | 67.28  | 20.53     | 54.98     |

(c) 4-shot

|           | Real  | Painting | Clipart | Sketch | Quickdraw | Infograph |
|-----------|-------|----------|---------|--------|-----------|-----------|
| Real      | 88.52 | -        | -       | -      | -         | -         |
| Painting  | 86.93 | 72.48    | -       | -      | -         | -         |
| Clipart   | 87.57 | 71.38    | 76.25   | -      | -         | -         |
| Sketch    | 86.92 | 70.98    | 76.24   | 68.74  | -         | -         |
| Quickdraw | 86.49 | 68.92    | 75.21   | 67.50  | 25.67     | -         |
| Infograph | 86.94 | 69.42    | 75.72   | 67.49  | 22.98     | 55.73     |

(d) 8-shot

|           | Real  | Painting | Clipart | Sketch | Quickdraw | Infograph |
|-----------|-------|----------|---------|--------|-----------|-----------|
| Real      | 88.52 | -        | -       | -      | -         | -         |
| Painting  | 87.42 | 72.13    | -       | -      | -         | -         |
| Clipart   | 87.11 | 72.90    | 76.36   | -      | -         | -         |
| Sketch    | 86.76 | 71.65    | 76.54   | 68.59  | -         | -         |
| Quickdraw | 85.82 | 69.41    | 76.38   | 68.29  | 29.53     | -         |
| Infograph | 87.21 | 72.25    | 75.98   | 67.82  | 27.15     | 56.84     |

Table 19: **Performance change across varying levels of supervision for CDDB-Hard dataset**, with seed value=2.

(a) 1-shot

|  | GauGAN | BigGAN | Wild | WhichfaceisReal | SAN |
|---|---|---|---|---|---|
| GauGAN | 98.90 | - | - | - | - |
| BigGAN | 89.15 | 66.75 | - | - | - |
| Wild | 69.35 | 68.12 | 45.95 | - | - |
| WhichfaceisReal | 51.00 | 53.00 | 51.04 | 50.25 | - |
| SAN | 53.75 | 56.87 | 52.30 | 50.25 | 63.85 |

(b) 2-shot

|  | GauGAN | BigGAN | Wild | WhichfaceisReal | SAN |
|---|---|---|---|---|---|
| GauGAN | 98.85 | - | - | - | - |
| BigGAN | 91.40 | 91.37 | - | - | - |
| Wild | 79.60 | 76.87 | 55.21 | - | - |
| WhichfaceisReal | 80.05 | 83.87 | 55.45 | 67.75 | - |
| SAN | 63.65 | 47.37 | 51.09 | 42.00 | 45.78 |

(c) 4-shot

|  | GauGAN | BigGAN | Wild | WhichfaceisReal | SAN |
|---|---|---|---|---|---|
| GauGAN | 98.80 | - | - | - | - |
| BigGAN | 96.15 | 79.62 | - | - | - |
| Wild | 90.70 | 84.37 | 64.03 | - | - |
| WhichfaceisReal | 85.60 | 86.00 | 64.42 | 80.25 | - |
| SAN | 90.45 | 86.12 | 56.51 | 70.50 | 61.44 |

(d) 8-shot

|  | GauGAN | BigGAN | Wild | WhichfaceisReal | SAN |
|---|---|---|---|---|---|
| GauGAN | 98.95 | - | - | - | - |
| BigGAN | 95.35 | 90.50 | - | - | - |
| Wild | 86.55 | 83.12 | 63.06 | - | - |
| WhichfaceisReal | 87.95 | 87.12 | 62.43 | 81.50 | - |
| SAN | 82.50 | 82.87 | 65.87 | 74.25 | 54.21 |

(e) 16-shot

|  | GauGAN | BigGAN | Wild | WhichfaceisReal | SAN |
|---|---|---|---|---|---|
| GauGAN | 98.95 | - | - | - | - |
| BigGAN | 94.60 | 94.50 | - | - | - |
| Wild | 74.95 | 84.00 | 61.85 | - | - |
| WhichfaceisReal | 93.60 | 88.50 | 59.91 | 83.00 | - |
| SAN | 93.90 | 88.75 | 63.59 | 83.25 | 59.03 |

Table 20: **Performance change across varying levels of supervision for CoRE50 dataset** with seed value = 2.

(a) 1-shot

|    | s1 | s2 | s3 | s4 | s5 | s6 | s7 | s8 |
|----|----|----|----|----|----|----|----|----|
| s1 | 98.00 | - | - | - | - | - | - | - |
| s2 | 91.70 | 80.93 | - | - | - | - | - | - |
| s3 | 92.50 | 81.83 | 80.86 | - | - | - | - | - |
| s4 | 92.30 | 80.76 | 79.67 | 82.93 | - | - | - | - |
| s5 | 90.73 | 80.23 | 78.90 | 79.67 | 84.27 | - | - | - |
| s6 | 89.16 | 75.76 | 74.93 | 78.33 | 78.46 | 79.30 | - | - |
| s7 | 88.33 | 74.43 | 72.87 | 78.56 | 74.53 | 71.70 | 77.13 | - |
| s8 | 88.67 | 77.23 | 75.26 | 80.33 | 76.40 | 68.53 | 73.30 | 83.96 |

(b) 2-shot

|    | s1 | s2 | s3 | s4 | s5 | s6 | s7 | s8 |
|----|----|----|----|----|----|----|----|----|
| s1 | 98.00 | - | - | - | - | - | - | - |
| s2 | 95.00 | 82.20 | - | - | - | - | - | - |
| s3 | 95.00 | 82.33 | 74.80 | - | - | - | - | - |
| s4 | 93.33 | 80.90 | 80.00 | 87.50 | - | - | - | - |
| s5 | 94.06 | 79.17 | 77.27 | 80.40 | 86.30 | - | - | - |
| s6 | 91.00 | 77.27 | 73.96 | 79.63 | 81.50 | 82.53 | - | - |
| s7 | 91.23 | 80.57 | 75.97 | 81.37 | 79.63 | 80.86 | 80.76 | - |
| s8 | 91.56 | 80.50 | 79.03 | 82.67 | 81.50 | 78.60 | 80.17 | 86.60 |

(c) 4-shot

|    | s1 | s2 | s3 | s4 | s5 | s6 | s7 | s8 |
|----|----|----|----|----|----|----|----|----|
| s1 | 98.00 | - | - | - | - | - | - | - |
| s2 | 92.63 | 80.13 | - | - | - | - | - | - |
| s3 | 93.20 | 81.33 | 84.03 | - | - | - | - | - |
| s4 | 93.50 | 79.76 | 82.56 | 87.96 | - | - | - | - |
| s5 | 94.96 | 79.63 | 80.30 | 82.40 | 86.93 | - | - | - |
| s6 | 93.43 | 78.13 | 78.00 | 82.80 | 80.70 | 83.20 | - | - |
| s7 | 94.13 | 81.36 | 78.60 | 84.93 | 83.13 | 80.00 | 82.90 | - |
| s8 | 93.77 | 82.80 | 80.00 | 83.66 | 83.36 | 78.13 | 80.43 | 88.60 |

(d) 8-shot

|    | s1 | s2 | s3 | s4 | s5 | s6 | s7 | s8 |
|----|----|----|----|----|----|----|----|----|
| s1 | 98.00 | - | - | - | - | - | - | - |
| s2 | 92.13 | 87.27 | - | - | - | - | - | - |
| s3 | 92.87 | 85.67 | 87.57 | - | - | - | - | - |
| s4 | 94.07 | 84.50 | 84.40 | 90.27 | - | - | - | - |
| s5 | 94.20 | 84.20 | 83.10 | 86.50 | 91.20 | - | - | - |
| s6 | 92.37 | 83.40 | 78.33 | 85.37 | 85.03 | 88.57 | - | - |
| s7 | 93.40 | 84.47 | 82.47 | 85.80 | 85.33 | 83.33 | 86.53 | - |
| s8 | 92.57 | 85.77 | 82.87 | 86.27 | 85.20 | 80.37 | 82.87 | 91.23 |

(e) 16-shot

|    | s1 | s2 | s3 | s4 | s5 | s6 | s7 | s8 |
|----|----|----|----|----|----|----|----|----|
| s1 | 96.47 | - | - | - | - | - | - | - |
| s2 | 78.23 | 76.87 | - | - | - | - | - | - |
| s3 | 87.13 | 81.03 | 84.63 | - | - | - | - | - |
| s4 | 88.43 | 81.50 | 80.60 | 88.53 | - | - | - | - |
| s5 | 88.23 | 82.10 | 79.33 | 84.90 | 90.80 | - | - | - |
| s6 | 89.23 | 81.33 | 75.40 | 81.70 | 85.77 | 87.30 | - | - |
| s7 | 88.80 | 82.20 | 77.57 | 81.77 | 83.97 | 82.07 | 86.07 | - |
| s8 | 89.60 | 83.03 | 80.20 | 85.47 | 85.67 | 79.73 | 81.90 | 90.47 |

# O   Prompt depth

We experiment with the depth of encoder blocks that we synergize with the proposed Encoder synergy module. We experiment with $J = 1, 3, 5, 9$ and $11$. As we see in Table 21, as we increase the depth of the Encoder synergy module, we see the performance increases, which indicates that the model is able to use CLIP pre-trained knowledge better, and hence, representation is learned and better stability of performance. But as we go to the last block, where the features are already mature, we see a dip in performance, which aligns with observations made in Khattak et al. (2022).

Table 21: Encoder synergy depth ablation

| Shots | $J = 1$ | | $J = 3$ | | $J = 5$ | | $J = 9$ | | $J = 11$ | |
|---|---|---|---|---|---|---|---|---|---|---|
| | **AA\***($\uparrow$) | **FA\***($\uparrow$) | **AA\***($\uparrow$) | **FA\***($\uparrow$) | **AA\***($\uparrow$) | **FA\***($\uparrow$) | **AA\***($\uparrow$) | **FA\***($\uparrow$) | **AA\***($\uparrow$) | **FA\***($\uparrow$) |
| **1-shot** | 73.22 | 62.58 | 72.32 | 60.71 | 73.10 | 62.23 | **74.27** | **63.76** | 73.34 | 63.12 |
| **2-shot** | 73.39 | 62.39 | 72.95 | 61.76 | 73.67 | 63.02 | **74.36** | **63.92** | 73.45 | 63.23 |
| **4-shot** | 73.58 | 62.65 | 73.53 | 62.56 | 73.83 | 63.30 | **74.93** | **64.48** | 73.49 | 63.45 |
| **8-shot** | 73.67 | 62.73 | 73.58 | 62.61 | 74.01 | 63.32 | **75.51** | **66.93** | 73.92 | 63.87 |

# P   Prompt Length ablation

Prompt length is an important hyperparameter of PGO-BEn. We detail the changes in the figure 8

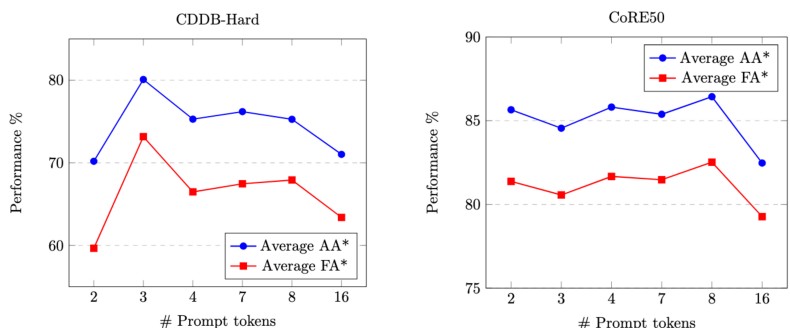

Figure 8: **Prompt length v/s AA\* and FA\* measure** by taking average of $1, 2, 4$ and 8-shots.

As we observe with very high prompt length, the performance dips.

# Q   Novel classes during inference

We experiment the scenario where the model encounters novel classes during inference time, and compare the results in Table 22. This is a very practical scenario in different use cases. Following the experimental setup of Khattak et al. (2022); Zhou et al. (2022b;a), we separate the set of classes of every domain into two groups, Base and New. During training, the model observes base classes, and we evaluate the performance on base classes and new classes.

We implemented the LwF, InfLORA and our method in this experiment with DomainNet dataset in 1-shot and 4-shot, where we learn the context vectors in the base class and for inference, we change the [CLS] token in the prompt $Pr$, to perform inference on new classes. Our method achieves superior stability as compared to other baselines, highlighting the superiority of our method and the applicability of our method in real-world scenarios where we can come across new classes after deployment, like autonomous driving.

Table 22: Comparing performance of PGO-BEN with two baseline methods (LwF, InfLORA) towards recognizing novel classes during inference.

| Shots | Method | Base | | New |
|---|---|---|---|---|
| | | **AA\*(↑)** | **FA\*(↑)** | **AA\*(↑)** |
| 1-shot | LwF\* | 72.04 | 66.77 | 70.29 |
| | InfLORA | 70.77 | 63.70 | 68.76 |
| | **PGO-Ben** | **77.74** | **68.51** | **76.00** |
| 4-shot | LwF\* | 71.79 | 69.04 | 70.10 |
| | InfLORA | 68.87 | 64.93 | 67.51 |
| | **PGO-Ben** | **78.88** | **69.58** | **77.08** |

# R   Experiments with more seed values

We discuss the results with more number of seed values in this section. The average of $1, 2, 4$ and 8-shot performance across five different seeds are detailed in Table 23. The results for DomainNet dataset is in Table 24, CDDB-Hard results in Table 25 and CoRE50 results in Table 26

Table 23: **Comparison across DomainNet, CDDB-Hard, and CoRE50 averaged over 1, 2, 4, and 8-shot settings**. **Bold** and underlined denote the best and second-best scores. PGO-BEN outperforms all baselines without using prompt pools, demonstrating its generalization strength. \* indicates CLIP-ViTB/16-based reimplementation. Results are mean ± std over 5 seeds. Red font denotes least std method.

| Method | Prompt Pool | Backbone | DomainNet | | CDDB-Hard | | CoRE50 | |
|---|---|---|---|---|---|---|---|---|
| | | | Average | | Average | | Average | |
| | | | **AA\*(↑)** | **FA\*(↑)** | **AA\*(↑)** | **FA\*(↑)** | **AA\*(↑)** | **FA\*(↑)** |
| DyTox Douillard et al. (2021) | × | ViT | $31.66_{\pm0.92}$ | $19.74_{\pm0.69}$ | $57.24_{\pm0.66}$ | $53.54_{\pm0.93}$ | $47.19_{\pm0.79}$ | $29.49_{\pm1.04}$ |
| Zero-shot CLIP Radford et al. (2021) | × | CLIP | 69.05 | – | 56.32 | – | 12.67 | – |
| LwF\* Li & Hoiem (2017) | × | CLIP | $72.21_{\pm0.86}$ | $60.77_{\pm0.96}$ | $68.06_{\pm0.83}$ | $58.85_{\pm0.82}$ | $64.80_{\pm0.86}$ | $57.75_{\pm0.66}$ |
| EwC\* Kirkpatrick et al. (2017) | × | " | $70.88_{\pm0.94}$ | $59.12_{\pm0.87}$ | $70.99_{\pm0.91}$ | $62.25_{\pm0.88}$ | $63.57_{\pm0.72}$ | $55.54_{\pm0.48}$ |
| L2P\* Wang et al. (2022c) | ✓ | " | $67.29_{\pm0.64}$ | $54.70_{\pm0.49}$ | $71.45_{\pm1.04}$ | $64.03_{\pm0.94}$ | $80.17_{\pm0.94}$ | $78.44_{\pm0.97}$ |
| DualPrompt\* Wang et al. (2022b) | ✓ | " | $73.51_{\pm0.64}$ | $63.50_{\pm0.74}$ | $72.85_{\pm0.74}$ | $66.46_{\pm0.90}$ | $55.87_{\pm0.59}$ | $50.89_{\pm0.74}$ |
| S-Prompt Wang et al. (2022a) | ✓ | " | $67.84_{\pm0.43}$ | $56.27_{\pm0.33}$ | $65.60_{\pm0.61}$ | $60.80_{\pm0.59}$ | $79.44_{\pm0.84}$ | $76.29_{\pm0.55}$ |
| CODA-Prompt Smith et al. (2023) | ✓ | " | $73.67_{\pm0.81}$ | $63.50_{\pm0.68}$ | $70.58_{\pm0.57}$ | $60.46_{\pm0.54}$ | $57.14_{\pm0.93}$ | $43.79_{\pm0.86}$ |
| InfLORA\* Liang & Li (2024) | × | " | $72.15_{\pm0.70}$ | $60.70_{\pm0.80}$ | $67.03_{\pm0.54}$ | $57.41_{\pm0.86}$ | $65.15_{\pm0.97}$ | $58.15_{\pm0.76}$ |
| CP-Prompt Feng et al. (2024) | ✓ | " | $72.19_{\pm0.85}$ | $61.13_{\pm0.77}$ | $66.88_{\pm0.41}$ | $62.21_{\pm0.25}$ | $81.68_{\pm0.70}$ | $79.96_{\pm0.59}$ |
| PGO-BEN **(Ours)** | × | CLIP | $\mathbf{74.85}_{\pm0.24}$ | $\mathbf{64.92}_{\pm0.30}$ | $\mathbf{79.69}_{\pm0.33}$ | $\mathbf{72.61}_{\pm0.19}$ | $\mathbf{86.38}_{\pm0.39}$ | $\mathbf{82.52}_{\pm0.57}$ |
| | | Δ | +1.18 | +1.42 | +6.84 | +6.15 | +4.70 | +2.56 |

As we can see, there are no changes to the relative ordering of the baseline methods with us, with our method clearly superior across all scenarios. The previous table with 3 seeds is mentioned in Table 2

Table 24: **Comparison with existing DIL benchmarks on DomainNet dataset across** 1, 2, 4 **and 8-shots.** We report the **AA\*** and **FA\*** values for comparison, which indicate 'Overall Average Accuracy' and 'Overall Forgetting Alleviation.' The highest performance is shown in **bold**, with the second highest underlined. PGO-BEN performs superior compared to all the baselines across the varying levels of supervision, highlighting the effectiveness of the proposed methodology.\* indicates reimplemented with CLIP. Results are mean $\pm$ std of 5 seeds.

| Method | Prompt pool | Backbone | 1-shot | | 2-shot | | 4-shot | | 8-shot | |
|---|---|---|---|---|---|---|---|---|---|---|
| | | | AA\*(↑) | FA\*(↑) | AA\*(↑) | FA\*(↑) | AA\*(↑) | FA\*(↑) | AA\*(↑) | FA\*(↑) |
| DyTox Douillard et al. (2021) | × | ViT | $30.21_{\pm0.73}$ | $18.96_{\pm0.66}$ | $29.68_{\pm0.91}$ | $18.86_{\pm0.81}$ | $36.16_{\pm1.12}$ | $22.59_{\pm0.62}$ | $30.61_{\pm0.95}$ | $18.58_{\pm0.67}$ |
| LwF\* Li & Hoiem (2017) | × | CLIP | $72.43_{\pm0.59}$ | $61.34_{\pm1.21}$ | $72.45_{\pm0.76}$ | $61.31_{\pm0.73}$ | $71.82_{\pm0.76}$ | $60.47_{\pm1.18}$ | $72.17_{\pm1.35}$ | $59.99_{\pm0.75}$ |
| EwC\* Kirkpatrick et al. (2017) | × | " | $71.83_{\pm1.18}$ | $61.03_{\pm0.99}$ | $71.02_{\pm0.62}$ | $59.87_{\pm0.85}$ | $70.41_{\pm0.95}$ | $57.89_{\pm1.21}$ | $70.26_{\pm1.01}$ | $57.71_{\pm0.45}$ |
| L2P\* Wang et al. (2022c) | ✓ | " | $65.59_{\pm0.58}$ | $53.15_{\pm0.32}$ | $67.36_{\pm0.76}$ | $55.26_{\pm0.93}$ | $67.97_{\pm0.66}$ | $54.98_{\pm0.41}$ | $68.24_{\pm0.58}$ | $55.41_{\pm0.31}$ |
| DualPrompt\* Wang et al. (2022b) | ✓ | " | $72.77_{\pm0.59}$ | $62.35_{\pm0.68}$ | $73.21_{\pm0.81}$ | $63.19_{\pm0.67}$ | $73.67_{\pm0.72}$ | $64.28_{\pm0.71}$ | $74.40_{\pm0.45}$ | $64.19_{\pm0.91}$ |
| S-Prompt Wang et al. (2022a) | ✓ | " | $62.66_{\pm0.39}$ | $50.55_{\pm0.41}$ | $67.82_{\pm0.57}$ | $55.95_{\pm0.47}$ | $69.83_{\pm0.20}$ | $58.68_{\pm0.19}$ | $71.05_{\pm0.58}$ | $59.92_{\pm0.28}$ |
| CODA-Prompt Smith et al. (2023) | ✓ | " | $\underline{72.98}_{\pm0.86}$ | $62.91_{\pm0.97}$ | $\underline{73.55}_{\pm0.72}$ | $62.99_{\pm0.73}$ | $\underline{73.68}_{\pm0.71}$ | $63.41_{\pm0.67}$ | $\underline{74.49}_{\pm0.97}$ | $64.72_{\pm0.36}$ |
| InfLORA\* Liang & Li (2024) | × | " | $72.83_{\pm0.61}$ | $61.67_{\pm0.68}$ | $72.11_{\pm0.87}$ | $61.09_{\pm0.79}$ | $71.53_{\pm0.72}$ | $60.68_{\pm1.02}$ | $72.03_{\pm0.62}$ | $59.36_{\pm0.72}$ |
| CP-Prompt Feng et al. (2024) | ✓ | " | $70.88_{\pm0.81}$ | $58.79_{\pm0.62}$ | $71.23_{\pm0.93}$ | $60.86_{\pm0.70}$ | $72.68_{\pm0.93}$ | $62.01_{\pm0.90}$ | $73.97_{\pm0.91}$ | $62.87_{\pm0.87}$ |
| PGO-BEN (Ours) | × | CLIP | $\mathbf{74.14}_{\pm0.21}$ | $\mathbf{63.98}_{\pm0.28}$ | $\mathbf{74.68}_{\pm0.35}$ | $\mathbf{64.12}_{\pm0.34}$ | $\mathbf{74.95}_{\pm0.17}$ | $\mathbf{64.71}_{\pm0.34}$ | $\mathbf{75.63}_{\pm0.23}$ | $\mathbf{66.90}_{\pm0.24}$ |
| Δ | | | +1.16 | +1.07 | +1.13 | +0.93 | +1.27 | +0.43 | +1.14 | +2.18 |

Table 25: **Comparison with existing DIL benchmarks on CDDB-Hard dataset across** 1, 2, 4 **and 8-shots.** We report the **AA\*** and **FA\*** values for comparison, which indicate 'Overall Average Accuracy' and 'Overall Forgetting Alleviation.' The highest performance is shown in **bold**, with the second highest underlined. PGO-BEN performs superior compared to all the baselines across the varying levels of supervision, highlighting the effectiveness of the proposed methodology.* indicates reimplemented with CLIP. Results are mean ± std of 5 seeds.

| Method | Prompt pool | Backbone | 1-shot | | 2-shot | | 4-shot | | 8-shot | |
|---|---|---|---|---|---|---|---|---|---|---|
| | | | AA*($\uparrow$) | FA*($\uparrow$) | AA*($\uparrow$) | FA*($\uparrow$) | AA*($\uparrow$) | FA*($\uparrow$) | AA*($\uparrow$) | FA*($\uparrow$) |
| DyTox Douillard et al. (2021) | × | ViT | $59.72_{\pm0.54}$ | $55.94_{\pm1.07}$ | $57.11_{\pm1.33}$ | $54.12_{\pm0.78}$ | $56.71_{\pm0.28}$ | $51.02_{\pm0.99}$ | $55.43_{\pm0.52}$ | $53.11_{\pm0.89}$ |
| LwF* Li & Hoiem (2017) | × | CLIP | $63.89_{\pm1.03}$ | $54.74_{\pm0.62}$ | $66.41_{\pm1.02}$ | $58.22_{\pm1.08}$ | $71.92_{\pm0.53}$ | $62.10_{\pm0.95}$ | $70.02_{\pm0.75}$ | $60.34_{\pm0.64}$ |
| EwC* Kirkpatrick et al. (2017) | × | " | $62.91_{\pm0.78}$ | $54.31_{\pm0.44}$ | $66.48_{\pm1.34}$ | $55.21_{\pm1.11}$ | $77.06_{\pm1.01}$ | $71.48_{\pm0.92}$ | $77.53_{\pm0.53}$ | $68.02_{\pm1.06}$ |
| L2P* Wang et al. (2022c) | ✓ | " | $65.31_{\pm0.74}$ | $57.43_{\pm1.11}$ | $73.92_{\pm1.41}$ | $67.31_{\pm0.78}$ | $74.11_{\pm0.97}$ | $64.88_{\pm1.01}$ | $72.48_{\pm1.04}$ | $66.51_{\pm0.88}$ |
| DualPrompt* Wang et al. (2022b) | ✓ | " | $71.41_{\pm0.66}$ | $65.82_{\pm0.75}$ | $73.19_{\pm0.52}$ | $65.01_{\pm0.78}$ | $72.49_{\pm0.99}$ | $68.29_{\pm1.12}$ | $74.34_{\pm0.81}$ | $66.72_{\pm0.97}$ |
| S-Prompt Wang et al. (2022a) | ✓ | " | $62.94_{\pm0.51}$ | $59.12_{\pm0.42}$ | $64.92_{\pm0.73}$ | $60.14_{\pm0.68}$ | $66.03_{\pm0.89}$ | $61.89_{\pm0.94}$ | $68.52_{\pm0.32}$ | $62.06_{\pm0.34}$ |
| CODA-Prompt Smith et al. (2023) | ✓ | " | $70.14_{\pm0.54}$ | $61.72_{\pm0.77}$ | $72.01_{\pm0.41}$ | $61.41_{\pm0.43}$ | $69.71_{\pm0.67}$ | $59.91_{\pm0.34}$ | $70.48_{\pm0.68}$ | $58.82_{\pm0.64}$ |
| InfLORA* Liang & Li (2024) | × | " | $63.39_{\pm0.34}$ | $53.71_{\pm0.92}$ | $62.34_{\pm0.71}$ | $54.22_{\pm0.72}$ | $69.48_{\pm0.54}$ | $59.26_{\pm0.74}$ | $72.94_{\pm0.57}$ | $62.48_{\pm1.08}$ |
| CP-Prompt Feng et al. (2024) | ✓ | " | $67.42_{\pm0.69}$ | $60.33_{\pm0.41}$ | $66.11_{\pm0.52}$ | $63.42_{\pm0.29}$ | $67.51_{\pm0.13}$ | $62.11_{\pm0.19}$ | $66.48_{\pm0.33}$ | $63.01_{\pm0.13}$ |
| PGO-BEN (Ours) | × | CLIP | $\mathbf{74.11_{\pm0.31}}$ | $\mathbf{66.42_{\pm0.17}}$ | $\mathbf{77.14_{\pm0.33}}$ | $\mathbf{71.12_{\pm0.25}}$ | $\mathbf{82.89_{\pm0.45}}$ | $\mathbf{75.91_{\pm0.12}}$ | $\mathbf{84.62_{\pm0.24}}$ | $\mathbf{77.02_{\pm0.22}}$ |
| | | Δ | +2.70 | +0.60 | +3.22 | +3.81 | +5.83 | +4.43 | +7.09 | +9.00 |

Table 26: **Comparison with existing DIL benchmarks on CoRE50 dataset across 1, 2, 4 and 8-shots.** We report the **AA\*** and **FA\*** values for comparison, which indicate 'Overall Average Accuracy' and 'Overall Forgetting Alleviation.' The highest performance is shown in **bold**, with the second highest underlined. PGO-BEN performs superior compared to all the baselines across the varying levels of supervision, highlighting the effectiveness of the proposed methodology.* indicates reimplemented with CLIP. Results are mean $\pm$ std of 5 seeds.

| Method | Prompt pool | Backbone | 1-shot AA*($\uparrow$) | 1-shot FA*($\uparrow$) | 2-shot AA*($\uparrow$) | 2-shot FA*($\uparrow$) | 4-shot AA*($\uparrow$) | 4-shot FA*($\uparrow$) | 8-shot AA*($\uparrow$) | 8-shot FA*($\uparrow$) |
|---|---|---|---|---|---|---|---|---|---|---|
| DyTox Douillard et al. (2021) | × | ViT | $45.82_{\pm0.60}$ | $27.86_{\pm1.28}$ | $48.24_{\pm1.38}$ | $30.67_{\pm1.18}$ | $45.78_{\pm0.98}$ | $27.98_{\pm1.05}$ | $48.95_{\pm0.74}$ | $31.45_{\pm0.68}$ |
| LwF* Li & Hoiem (2017) | × | CLIP | $59.95_{\pm0.77}$ | $53.09_{\pm0.33}$ | $66.45_{\pm0.76}$ | $57.31_{\pm0.66}$ | $64.92_{\pm0.94}$ | $57.21_{\pm0.90}$ | $67.89_{\pm0.98}$ | $63.40_{\pm0.76}$ |
| EwC* Kirkpatrick et al. (2017) | × | " | $58.67_{\pm0.90}$ | $51.99_{\pm0.61}$ | $66.38_{\pm0.30}$ | $58.11_{\pm0.52}$ | $63.84_{\pm0.80}$ | $54.83_{\pm0.30}$ | $65.41_{\pm0.91}$ | $57.24_{\pm0.50}$ |
| L2P* Wang et al. (2022c) | ✓ | " | $81.45_{\pm0.99}$ | $77.85_{\pm0.85}$ | $80.94_{\pm0.96}$ | $79.88_{\pm0.93}$ | $79.32_{\pm0.83}$ | $77.98_{\pm1.33}$ | $79.00_{\pm0.98}$ | $78.06_{\pm0.78}$ |
| DualPrompt* Wang et al. (2022b) | ✓ | " | $44.10_{\pm0.58}$ | $36.78_{\pm0.88}$ | $59.80_{\pm0.53}$ | $55.62_{\pm1.02}$ | $58.67_{\pm0.31}$ | $52.72_{\pm0.28}$ | $60.92_{\pm0.94}$ | $58.46_{\pm0.81}$ |
| S-Prompt Wang et al. (2022a) | ✓ | " | $77.84_{\pm0.85}$ | $74.18_{\pm0.60}$ | $79.94_{\pm0.90}$ | $74.88_{\pm0.32}$ | $79.72_{\pm0.76}$ | $77.34_{\pm0.61}$ | $80.28_{\pm0.87}$ | $78.79_{\pm0.70}$ |
| CODA-Prompt Smith et al. (2023) | ✓ | " | $54.25_{\pm0.87}$ | $40.61_{\pm0.92}$ | $55.86_{\pm0.73}$ | $41.62_{\pm0.63}$ | $57.96_{\pm0.80}$ | $44.80_{\pm0.90}$ | $60.52_{\pm1.34}$ | $48.15_{\pm1.01}$ |
| InfLORA* Liang & Li (2024) | × | " | $57.18_{\pm1.05}$ | $50.44_{\pm0.82}$ | $66.46_{\pm1.12}$ | $57.72_{\pm0.90}$ | $63.78_{\pm0.67}$ | $58.21_{\pm0.50}$ | $73.18_{\pm1.06}$ | $66.24_{\pm0.82}$ |
| CP-Prompt Feng et al. (2024) | ✓ | " | $78.40_{\pm0.58}$ | $77.28_{\pm0.42}$ | $81.78_{\pm0.30}$ | $78.65_{\pm0.60}$ | $82.40_{\pm1.18}$ | $81.19_{\pm1.04}$ | $84.17_{\pm0.75}$ | $82.72_{\pm0.30}$ |
| PGO-BEN (Ours ) | × | CLIP | $\mathbf{83.11}_{\pm0.31}$ | $\mathbf{78.22}_{\pm0.39}$ | $\mathbf{86.64}_{\pm0.37}$ | $\mathbf{82.75}_{\pm0.32}$ | $\mathbf{87.34}_{\pm0.52}$ | $\mathbf{83.77}_{\pm0.92}$ | $\mathbf{88.46}_{\pm0.39}$ | $\mathbf{85.37}_{\pm0.67}$ |
| | | Δ | +1.66 | +0.37 | +4.86 | +2.87 | +4.94 | +2.58 | +4.29 | +2.65 |

