# OpenReview forum: "\textsc{PGO-BEN}: Proxy-Guided Orthogonalization and Beta Ensembling for Few-Shot Domain-Incremental Learning"
_TMLR — Accepted by TMLR_

### Review · Reviewer_R6Eh · 2025-11-01

**Summary Of Contributions:**

This paper introduces a new problem: Few-Shot Domain-Incremental Learning (FSDIL). It differs from prior continual learning settings (such as DIL or FSCIL) as it addresses the practical challenge of adapting models to a sequence of new domains using only a few labeled samples, without access to past data.

To solve this problem, the paper proposes a novel learning method: PGO-BEn. Comparing to previous work, this method has the following novelties:
1. Proxy-Guided Orthogonalization (PGO): Unlike prior gradient project methods that require storing past gradient/data, PGO uses the current model (from last session) as a proxy for past knowledge and projects conflicting updates into an orthogonal subspace to prevent forgetting.
2. Beta Ensemble (BEn): Instead of the standard Exponential Moving Average (EMA), the paper proposes an ensemble strategy that leverages a Beta distribution to adaptively balance weights between early model states (preserving old knowledge) and latest model states (adapting to the new distribution).

The paper conducted extensive evaluations to show the efficacy of the proposed method PGO-BEn. Various ablation study are also conducted to validate the contribution of the proposed components.

**Audience:**

Yes

**Audience Explanation:**

The problem investigated in this paper, FSDIL, in which a model needs to adapt to a sequence of new domains using only a few labeled samples per class, with the constraint of being "rehearsal-free", is interesting and differs from prior work. The paper provides a benchmark datasets and the related metrics for evaluation.

The proposed PGO-BEn is a novel yet effective method for this specific problem. In particular, the Beta moving average is a simple but quite effective method to preserve prior (and the base) model knowledge, which might be very useful in similar settings (like generic CL).

**Claims And Evidence:**

Yes

**Claims Explanation:**

1. The introduced problem FSDIL is different from prior challenges such as DIL and FSCIL, and the authors clearly discussed their differences.
2. The proposed method has two major contributions: PGO and BEn. The similarity and differences between prior arts are discussed in the paper. The efficacy of the proposed method are validated in the main results (Table 2) and the ablation study (4.2b, 4.2d, and 4.2e).

**Requested Changes:**

The graphical representations in this paper contain non-standard scaling choices that potentially distort the interpretation of the data. In Figure 1, the radar graph's visual magnitude is compromised by a non-zero or inconsistent origin point, causing the result of 20.84 to appear disproportionately larger than 68.77. Furthermore, Figures 4, 5, and 6 employ a truncated Y-axis (non-zero baseline), which exaggerates the perceived magnitude of small differences. Specifically, Figure 5 (Real -> Sketch) displays values of 32.06 and 31.95 (a difference of 0.11), yet the Y-axis begins at 31.9, visually inflating the modest performance gap.

To maintain clarity and strengthen the communication, I would like the authors to update the figures to use standard axes.

---

> ### Author Response · Authors · 2025-11-27
>
> We sincerely thank the reviewer for the suggestion.
>
>
> We have updated the radar-plot in Fig. 1 and converted the Fig. 5 and 6. to Table 4 and 5. Fig. 4 is a pictorial depiction of the difference and exact data for 4b) is already present in Sup.Mat. Tab. 15.

---

### Review · Reviewer_icEg · 2025-11-10

**Summary Of Contributions:**

1.This paper addresses a realistic problem: pre-trained vision–language models often lack access to data from previous training tasks when performing few-shot domain-incremental learning (FSDIL), leading to catastrophic forgetting and limited plasticity. This challenge arises in multiple real-world applications, such as autonomous driving and medical imaging.

2.The paper proposes PGO-BEn (Proxy-Guided Orthogonalization and Beta Ensembling), a rehearsal-free framework for FSDIL. It consists of two key components:

(2.1) Proxy-Guided Orthogonalization (PGO) identifies conflicts between the current update and previously learned knowledge, and mitigates the conflicts by projecting the conflicting gradient into an orthogonal subspace, and

(2.2) Beta Ensembling (BEn) stabilizes training by maintaining a Beta Moving Average (BMA) of previous intermediate models.

PGO-BEn is supported by both theoretical and empirical evidence.

**Additional Comments:**

I would appreciate if the authors could explain more on the learnable prompts introduced across encoders. A learnable token may cause the distribution shift of inputs even in the same task. Would this shift be a cause of instability during learning, and if yes, how does PGO-BEn mitigate that?

**Audience:**

Yes

**Audience Explanation:**

The problem addressed in this work is relevant to real-world applications, particularly when training domains contain sensitive data or when memory resources are limited.

The paper considers the practical applicability in design choices. The rehearsal-free setup aligns with realistic constraints where previously seen data cannot be reused due to privacy or storage limitations, broadening the potential impact of the method. Moreover, Equation (5) presents an efficient online implementation of Beta Moving Average (BMA) that reduces memory overhead. Additionally, the orthogonal projection step in Equation (3) adds minimal computational cost without introducing notable concern regarding the Big-O analysis.

**Claims And Evidence:**

Yes

**Claims Explanation:**

Overall, the claims are clear and generally convincing, though certain experimental design choices limit the accuracy of the empirical evidence.

**Evidence of contribution 1:**

(a) The paper defines the Few-Shot Domain-Incremental Learning (FSDIL) problem clearly. Section 1 (the second paragraph on page 2) introduces the concept, and Figure 2(a) provides an overview of the pipeline. A formal definition follows at the beginning of Section 3 (page 4).

(b) The proposed model update procedure closely follows the FSDIL setting, relying solely on data from the current domain and previously learned model states, without requiring access to data from earlier domains. Equations 1–6 and the first equation on the top of page 6 confirm this property.

**Evidence of contribution 2:**

(a) The paper provides a step-by-step explanation of PGO-BEn in Section 3.1. The pseudocode is provided in Algorithm 1 in appendix J.

(b) The method is supported by a PAC-Bayesian theoretical analysis in Section 3.2, which establishes a risk bound for PGO-BEn.

(c) Figure 1 quantifies PGO-BEn’s performance across multiple domains, demonstrating improvements over a zero-shot CLIP baseline.

(d) Table 2 and Figure 4(b) empirically supports the performance of PGO-BEn on three benchmarks, DomainNet, CoRE50, and CDDB-Hard. The performance is quantified using two metrics: average accuracy and forgetting alleviation. PGO-BEn consistently outperforms baseline methods on both. The detailed metric definitions are provided in Appendix H.
(e) Figure 4(a) compares PGO-BEn’s representation alignment to baselines, highlighting that the proposed method learns more aligned representations than baselines.

(e) The paper conducts comprehensive ablation studies to support each component’s contribution.
Figure 5 demonstrates that BMA, a component of PGO-BEn, is more robust to large domain shifts compared to  exponential moving average (EMA).

Table 3 demonstrates BMA’s higher average accuracy and forgetting alleviation compared to EMA.

Table 5 suggests that PGO and BMA both contribute to the stability of the method. Using only one of them causes a performance degradation.

Table 6 empirically shows the advantage of the proposed multimodal setup in prompting configuration.

(f) Table 4 discusses sensitivity to a key parameter, analyzing performance under both lower and higher values.

(g) Detailed implementation information is included in Appendix I.

**Issue and limitation:**

A notable weakness lies in the accuracy of experimental results. Table 2 reports results using only three random seeds, which is insufficient to provide statistically stable mean and standard deviation (STD) estimates. Increasing the number of seeds would improve the accuracy of the reported numbers. Similarly, Figure 6 omits information about the number of seeds used, making it difficult to assess the reliability of the reported variance.

**Requested Changes:**

The empirical evidence could be made more accurate and convincing by using additional random seeds, particularly when reporting the average accuracy and forgetting alleviation. Additionally, reporting the number of seeds used in all experiments would improve reproducibility.

The letter $n$ in PGO-BEn is inconsistent, appearing in both lowercase and uppercase. It would be clearer to adopt a consistent format.

---

> ### Author Response · Authors · 2025-11-27
>
> RQ-C $\rightarrow$ Requested Change ,  AQ $\rightarrow$ Additional Query
>
> RQ-C-1) **Experiment with more seeds** Our reported results in paper, were average of 3 seeds. We have updated Table 2 and added experimental results with $5$ seed values in Sup.Mat. Section R (Table 23, 24, 25 for DomainNet, CDDB-Hard and CoRE50 respectively) on all the three datasets across all the shots. Original Table 2 shifted to Table 26 (section R Sup.Mat.)
>
>
> RQ-C-2) Thank you for pointing out. We have rectified the inconsistency of lowercase and uppercase of "n" in PGO-BEn.
>
> AQ 1) **Shift due to learnable prompt**:
>
> We thank the reviewer for the useful comment. We agree that a new learnable token may cause the distribution shift in the representation space of the CLIP image encoder. But we would like to highlight that such risk is very unlikely.
>
>
> *   Due to the Vision encoder backbone of ViT-B/16  (a Transformer based architecture), has very low inductive bias and with the entire CLIP backbone weight being frozen, the introduced prompt tokens doesn't cause drastic shift. Rather, such introduced prompt tokens across layers give the model flexibility to learn representations across layers.
> *   This also allows to learn a domain-agnostic representation which can then be shared with the Text-Encoder via the projector networks, to more suitably align the text-encoder representation to the changing visual domain. If we do not introduce any such learnable tokens, then to share the information with the Text encoder, we can take an average of the patch token representations and pass it to the Text encoder, via Projector networks. But such a design fails to capture overall data distribution, and gets biased to the. current batch of input data.
> *   Since the new tokens would be initialized in the base session itself, there are sufficient amount of data available for the model to tune those prompt token representation. The learned prompt on base session domain will be then used to initialize in the second session onwards. Our training mechanism of proxy-guided orthogonalization followed by beta-ensembling doesn't allow stark weight updates thus simultaneously mitigating the risk of such behaviour.
>
> We extend our experiments on the encoder-depth synergy module (from Table 21) with $J=0$. $J=0$ discusses the scenario of no prompt tokens being introduced in the encoders and the only learnable parameter is the prompt input to the Text-Encoder.
>
> |Method|1-shot AA*↑|1-shot FA*↑|2-shots AA*↑|2-shots FA*↑|
> |-|-|-|-|-|
> |$J=0$|71.34|61.22|71.77|61.36|
> |$J=1$|73.22|62.58|73.39|62.39|
> |$J=9$|74.27|63.76|74.36|63.92|

---

### Review · Reviewer_XRrb · 2025-11-19

**Summary Of Contributions:**

This paper introduces PGO-BEn, a framework designed for FSDIL, which leverages pre-trained CLIP models via (i) a novel multi-modal prompting strategy that conditions text prompts on vision prompts, (ii) a proxy-guided orthogonalization mechanism for regularizing gradient updates to preserve prior knowledge, and (iii) a Beta-function-based temporal ensembling technique that adaptively weights intermediate model states to mitigate forgetting.

**Audience:**

Yes

**Audience Explanation:**

FSCIL is an important issue in the ML community.

**Claims And Evidence:**

Yes

**Claims Explanation:**

1. Table 2 presents comprehensive results across three challenging datasets and four few-shot settings.
2. The paper includes a PAC-Bayesian analysis justifying the framework's design, as well as detailed mathematical explanations for the orthogonalization and ensembling components.

**Requested Changes:**

1. Can the authors provide empirical or qualitative comparisons to the most recent prompt-based or language-augmented FSCIL works (e.g., PL-FSCIL, Language-Inspired Relation Transfer), and clarify what is gained versus such approaches?
2. Non-prompt strategies, alternative ensembling, and distributional shift adaptation methods are omitted
3. The computational cost for beta-ensembling and gradient computations across many domains is not quantified.
4. What is the empirical/memory/computational overhead of PGO-BEn when N (the number of domains) or the prompt-token space grows much larger?

---

> ### Author Response · Authors · 2025-11-27
>
> RQ-C $\rightarrow$ Requested Change
>
> We would like to clarify that our proposed methodology addresses Few-Shot Domain Incremental Learning (FSDIL), not FSCIL. To the best of our knowledge, FSDIL is a new setting that has not been explored in prior work. The distinctions between FSDIL and existing continual learning paradigms such as Class-Incremental Learning (CIL) and Few-Shot Class-Incremental Learning (FSCIL) are summarized in Tab. 1.
>
> RQ-C-1) Like our proposed method, PL-FSCIL also doesn't require explicit domain label. It learns the domain prompt simultaneously from the data, regularizing it to be orthgonal to the task-specific prompts. We ran the code for PL-FSCIL on 1-shot and 2-shot on the DomainNet dataset.
>
> |Method|1-shot AA*↑|1-shot FA*↑|2-shots AA*↑|2-shots FA*↑|
> |-|-|-|-|-|
> |PL-FSCIL|54.51|42.49|56.67|43.58|
> |Ours|74.27|63.76|74.36|63.92|
>
> PL-FSCIL operates under a class-incremental setting, assuming new classes arrive in each session. It enforces the domain prompt to remain orthogonal to class-specific prompts, a constraint unsuitable for FSDIL where domains shift but class labels remain fixed. Moreover, PL-FSCIL embeds domain knowledge in a single hidden layer of the vision encoder, which is suboptimal since different layers capture complementary domain-specific cues. In contrast, our encoder-synergy module leverages layerwise knowledge throughout the encoder, enabling more effective domain adaptation and leading to superior performance over PL-FSCIL.
>
> Language-Inspired Relation Transfer (LRT), TPAMI-2025, relies on external textual data in the case of adapting to few-shot labeled samples in incremental sessions. Given that LRT was designed for FSCIL, they used a simple prompt of `a photo of a [CLS]` as the text prompt (for fair comparison of text-guided FSCIL benchmarks). Such prompt is feasible as only the class labels change. In our setting of Domain Incremental Learning, domains change across session, so the textual-prompt requires an explicit domain label information. We do not have prior knowledge of the textual domain label. Thus we feel comparison with LRT is not fair and doesn't align with our problem setup.
>
> RQ-C 2) **Non-Prompt based and distribution-shift adaptation methods** We would like to polietly highlight that we have compared with DyToX (in Table 2, for all datasets, for all shots) which is not a prompt-based approach. The remaining non-prompt based methods (EwC, LWF) are all reimplemented to have the same backbone and thus a fair comparison with our method.
>
> We also compare with a recent non-prompt based adaptation method called DuCT (CVPR 2025)
>
> |Method|1-shot AA*↑|1-shot FA*↑|2-shots AA*↑|2-shots FA*↑|
> |-|-|-|-|-|
> |DyToX|29.94|18.72|29.20|18.20|
> |DUCT|60.04|44.70|60.04|45.18|
> |Ours|74.27|63.76|74.36|63.92|
>
> We tried with several of intializations like DiNO, ImageNet21K, SAM. The reported results are the best among them (specifically ImageNet21K initialization).
>
> **Alternative Ensembling**
> We have discussed about EMA and BMA based ensembling techniques, giving time varying weightage to individual model states. Other alternative weightage is to give equal weights to each model state. This can be achieved by giving $\beta=(1, 1)$, as visualized in Fig. 6). We have detailed the ablation in Table 3 and 6. We discuss the 1-shot and 2-shot results here and request the reviewer to refer to Table 3 and 6 for detailed results.
>
> |Method|1-shot AA*↑|1-shot FA*↑|2-shots AA*↑|2-shots FA*↑|
> |-|-|-|-|-|
> |EMA($\lambda=0.98$)|64.93|49.53|72.77|61.28|
> |BMA($\beta=(0.5, 0.5)$)|74.27|63.76|74.36|63.92|
> |BMA($\beta=(1, 1)$)|74.17|63.74|74.32|63.86|
>
> RQ-C-3) **Quantify computation cost:**  The computational cost of gradient computation and applying beta-ensembling while adapting to domain $\mathcal{D}_t$ is constant w.r.t. t, for $t>1$. Memory usage doesn't grow with the number of encountered domains as for any domain $\mathcal{D}_t$ for $t>1$, we only need to maintain model state $\mathcal{M}_t$ and $\mathcal{M}_t-1$. On DomainNet dataset (batchsize=128) GPU usage was 20.53GB for t=1 and 25.10GB for t>1. All experiments were run on A6000 48GB GPU as already mentioned in Section I (Sup.Mat.)
>
> RQ-C-4) **Computation overhead for larger number of domains:** For any incremental domain after the base session, the adaptation cost remains constant and is independent of the number of previously encountered domains. At any given incremental domain (say $\mathcal{D}_t$), we only need two model states with us, model $\mathcal{M}_t$ and $\mathcal{M}_t-1$ (frozen). If we introduce more prompt tokens inside the encoders, the projecter network's parameter would not vary. But the number of learnable paramter would increase, due to increase in the number of prompt tokens. But that increase w.r.t. CLIP ViTB/16 total number of parameter is negligible. As an example, if we add one more prompt token across each encoder layer then the total increase would be approximately 0.0076\% w.r.t. to CLIP ViTB/16 model.

---

### Author Response · Authors · 2025-12-11
**Further queries?**

Dear Reviewers,

Thanks a lot for your constructive comments. Please let us know if any further clarification is required. We are looking forward to respond to the same.

Best regards,

Authors

---

### Decision · Action_Editor_URPh · 2025-12-23

**Recommendation:** Accept as is

**Audience:**

Yes

**Audience Explanation:**

Yes. The paper addresses a practically important and timely problem (few-shot domain-incremental learning under an exemplar-free/privacy-aware constraint) and proposes simple, generally applicable techniques validated on multiple benchmarks.

**Claims And Evidence:**

Yes

**Claims Explanation:**

Yes. The main claims are supported by clear evidence: the paper shows consistent improvements across multiple benchmarks and few-shot settings, provides ablations to verify the contribution of key components (PGO and BMA), and the rebuttal strengthens credibility by adding more random-seed runs and correcting potentially misleading visualizations.